# BEYOND ACCURACY: UNDERSTANDING THE PERFORMANCE OF LLMS ON EXAMS DESIGNED FOR HUMANS

## ABSTRACT

Many recent studies of LLM performance have focused on the ability of LLMs to achieve outcomes comparable to humans on academic and professional exams. However, it is not clear whether such studies shed light on the extent to which models show reasoning ability, and there is controversy about the significance and implications of such results. We seek to look more deeply into the question of how and whether the performance of LLMs on exams designed for humans reflects the true aptitude inherent in LLMs. We do so by making use of the tools of psychometrics which are designed to perform meaningful measurements in test taking. We leverage a unique dataset that captures the detailed performance of over 5M students across 8 college entrance exams given over a span of two years in Brazil. With respect to the evaluation of LLM abilities, we show that the tools of Item Response Theory (IRT) provide a more informative evaluation of model performance than the usual accuracy metrics employed in previous studies. Digging deeper, we show that the modeling framework of IRT, by explicitly modeling the difficulty levels of questions, allows us to quantitatively distinguish between LLMs that answer questions in "human-like" patterns versus LLMs that do not. We also show how to quantitatively identify cases in which exam results are not reliable measurements of an LLM's ability. Using the tools of IRT we can also identify specific questions that appear to be either much easier, or much harder, for machines than for humans, and we give some reasons for those differences. Overall, our study shows that the conventional focus on accuracy as the primary performance metric for LLM studies does not allow us to deeply understand the true capabilities of LLMs and compare them to that of humans. Thus, we claim that psychometric modeling should play a larger role in the evaluation of LLM capabilities on exams designed for humans.

## 1 INTRODUCTION

Large Language Models (LLMs) have demonstrated an impressive ability to perform well on examinations designed for humans (Narayanan & Kapoor, 2023; Raji et al., 2021), such as the US bar exam (OpenAI, 2023), the US Medical Licensing Exam (Kung et al., 2023), and many others (Varanasi, 2023; Zhong et al., 2023). This yields controversy in how researchers should interpret such results, raising two kinds of criticisms of those apparent successes. The first is the potential for publicly-given exams (and answers) to leak into models' training data. The second, and more fundamental, issue is the notion of *construct validity* (Trott, 2024). Most exams given to humans are intended to measure a construct, e.g., legal analysis ability, medical analysis ability, etc. However, the reliability of these exams in measuring the relevant construct for non-humans is usually ignored, and exams that are valid in one context may not generalize across different groups, settings or tasks (Messick, 1995).

Formalizing the notion of construct validity in general is challenging. Since the 1950s, the field of psychometrics has been grappling with how to design examinations that validly measure human abilities along specific dimensions. The primary tool developed has been Item Response Theory (IRT) (Chen et al., 2021), which has been employed in psychology, medicine, and especially in educational testing. IRT formalizes the unobserved construct as a continuous latent variable, and models stochastic responses of humans to questions as a logistic regression conditional on that latent variable.

In this paper, we demonstrate how IRT can help shed light on whether LLMs are in fact showing human-like performance on exams intended for humans. As a case study, we use one of the largest university entrance exams in the world, a dataset comprising the performance of over 5 million Brazilian students on eight multiple-choice exams administered over two years. Each exam was prepared and fitted to an IRT model by educational testing experts, giving us an unparalleled opportunity to examine the performance of LLMs in detail.

Our results show that the LLMs we study reveal performance patterns that are consistent with expected human behavior in many cases. Nonetheless, we also frequently observe significant deviation from human-like behavior. We demonstrate how to use the tools of IRT to quantitatively distinguish between human-like and non-human-like behavior. We then explore the differences between models and exam types that correlate with differences in response patterns. Lastly, we use the tools of IRT and psychometrics to identify cases where exams are not producing reliable estimates of LLM ability and understand why this happens. This occurs because exams are in some cases too difficult for the models, and in other cases too easy for them, and as such they cannot properly measure the ability of certain LLMs.

Moving beyond conclusions about current models, the broader contribution of our study is to demonstrate the power of IRT as a framework for evaluating LLMs. For example, in Classical Test Theory (CTT), no attempt is made to assess the difficulty of individual questions, in-line with the majority of standard LLM benchmarks that pursue accuracy (Bubeck et al., 2023; Touvron et al., 2023; Hendrycks et al., 2020; bench authors, 2023). In contrast, as we will show below, IRT simultaneously measures both test takers and exam questions (on the same scale). In doing so, IRT allows one to distinguish between test takers with similar CTT (accuracy) scores, but differing levels of true ability, by inspecting the *pattern* of correct or incorrect answers given. Moreover, we deploy a broader set of tools (e.g., goodness-of-fit, Fisher information, discrimination index) which enable us to evaluate which are the cases in which fitting the IRT model to the LLMs response patterns gives us reliable estimates of the models' ability. Thus, we believe that the methods of our study represent a valuable step beyond the use of simple accuracy for assessing whether both current and future LLMs show human-like response patterns.

## 2 RELATED WORK

Our study connects a number of research areas, spanning benchmarking LLMs, the applications of item response theory, and the evaluation of LLMs using exams designed for humans.

**Benchmarking LLMs.** The most common strategy to evaluate LLMs is through traditional large-scale NLP benchmarks (Wang et al., 2018; Talmor et al., 2019; Cobbe et al., 2021; Joshi et al., 2017; Hendrycks et al., 2020; Kiela et al., 2021; bench authors, 2023). Conventionally, benchmark evaluation relies on some notion of accuracy – the number of correct answers – as a proxy for ability (Bubeck et al., 2023; Touvron et al., 2023). A key distinction of our study is to draw attention to the limitations of the use of accuracy alone (Schaeffer et al., 2023) for evaluating the performance of LLMs on benchmarks in understanding the similarity between the performance of models versus humans.

**LLMs and Exams Designed for Humans.** Many attempts to evaluate LLMs use exams designed for humans, e.g., at college-entrance (Achiam et al., 2023; Nunes et al., 2023) or college-level (Drori et al., 2023; Silva et al., 2023; Wang et al., 2023; Drori et al., 2022; Terwiesch, 2023; Zhong et al., 2023). These exams also generally use accuracy as a metric of ability; one focus of our work is on how to use IRT analysis to determine when such exams in fact perform meaningful measurement.

The Brazilian nationwide college-entrance exams we use in this work (ENEM), detailed in Section 4.1, were used in previous efforts to evaluate NLP models (Silveira & Deratani Mauá, 2017; Silveira & Mauá, 2018; Nunes et al., 2023). However, those studies only used accuracy and did not make use of the IRT models associated with the exam, which is a central aspect our work.

**IRT in Machine Learning.** Work in psychometrics (i.e., the measurement of human cognitive abilities), detailed in Section 3, has shown that using accuracy as a exam score may not reflect the true underlying abilities of individuals (Erguven, 2013). As a result, IRT has been advocated for use in machine learning (ML) as an improved tool for benchmarking. The authors in (Rodriguez

et al., 2021) show that it is possible to produce rankings of NLP models which are more reliable and stable using IRT than accuracy. Item response theory has also been shown to help in spotting noisy questions, identifying overfitting, selecting features, and designing better and smaller benchmarks for ML (Polo et al., 2024; Plumed et al., 2016; Sedoc & Ungar, 2020; Kline et al., 2021; Zhuang et al., 2023; Lalor et al., 2016). However, there is a critical difference between the previous uses of IRT in ML and our work. Previous work uses IRT by training an IRT model on the results of ML models solving question-answering or classification questions. Our method is different: we leverage the fact that we have access to an IRT model trained on *human responses,* and we do not retrain on *model responses.* We take this approach because a central goal of our study is to explore whether LLMs are in fact following response patterns *as exhibited by human test takers.*

Finally, we note that (Tjuatja et al., 2024) shares some goals with our work. The investigation seeks to understand whether LLMs show human-like response biases in surveys. We also look at the question of whether LLMs show human-like response patterns, but we study the question along different dimensions: (a) patterns of correct and incorrect answers in exams; and (b) the ways in which LLMs choose incorrect answers. Additionally, Xia *et al.* (Xia et al., 2024) recognize that accuracy as a single metric does not capture errors LLMs can make in intermediate steps when solving mathematical tasks, and they systematically study those errors.

## 3 BACKGROUND

In this section, we give some background of the tools we use from psychometrics.

**Classical Test Theory (CTT):** CTT (Allen & Yen, 2002) evaluates test takers based on the fraction of questions they answer correctly. We call this score *accuracy* or *CTT* score of the test taker and we use these two terms interchangeably. Inadequately, CTT does not differentiate between difficult and easy questions, nor does it take into consideration the *patterns of correct answers*. For example, the CTT score does not penalize a test taker who answers correctly difficult questions, but answers wrongly easy ones – despite the fact that such a pattern might be indicative of randomness or cheating.

**Item Response Theory (IRT):** IRT (De Ayala, 2013; Baker & Kim, 2004) is a model used extensively in psychometrics to measure the ability level of the test takers and evaluate the difficulty of the test questions (which are referred to as *items* in psychometrics). IRT takes into consideration the difficulty of the questions when evaluating the test-taker's performance and also makes use of the pattern of correct and incorrect responses on the exam. The model associates with every test taker $j$ a parameter $\theta_j$, which corresponds to the *ability* of $j$. The two-parameter IRT model (2PL) associates every question $i$ with two parameters $\phi_i = (\alpha_i, \beta_i)$. The model assumes that a test taker with ability $\theta_j$ answers question $i$ associated with $\phi_i$ correctly with probability given by the logistic function:

$$p_{ij} = \frac{e^{\alpha_i(\theta_j - \beta_i)}}{1 + e^{\alpha_i(\theta_j - \beta_i)}}. \tag{1}$$

Parameter $\alpha_i$ is the *discrimination parameter* and $\beta_i$ is the *difficulty* of question $i$. Note that the ability $\theta_j$ and the difficulty level $\beta_i$ are in the same scale; after all, the difference $(\theta_j - \beta_i)$ directly affects $p_{ij}$. For fixed $\alpha_i$, the difficulty parameter $\beta_i$ is the value (on the ability scale) for which $p_{ij} = 0.5$. Parameter $\alpha_i$ characterizes how well question $i$ can differentiate among test takers located at different points of the ability continuum; $\alpha_i$ is proportional to the slope of $p_{ij} = p_i(\theta_j)$ at the point where $p_{ij} = 0.5$ – the steeper the slope, the higher the discriminatory power of $i$. All the parameters of this model take values in $(-\infty, +\infty)$. Note that any set of questions comprising an exam spans a certain range of $\beta_i$ values; such a set is not appropriate to assess test takers with abilities outside this range.

The 3-Parameter IRT model (3PL for short) is an extension of the above model that also incorporates a *pseudo-guessing parameter* $\gamma_i$. Thus, in 3PL every question $i$ is associated with three parameters $\Phi_i = (\alpha_i, \beta_i, \gamma_i)$; $\alpha_i$ and $\beta_i$ are the same as before. Intuitively, $\gamma_i$ is the probability of answering correctly based on a random guess with $\gamma_i \in [0, 1]$. Thus, the probability of a test taker with ability $\theta_j$ to answer question $i$ correctly is:

$$P_{ij} = \gamma_i + (1 - \gamma_i)p_{ij}.$$

Given test-taker responses, the parameters of the model can be estimated using Bayesian methods (Baker & Kim, 2004). In our case, the ENEM dataset came with a set of questions for which the parameters $(\alpha_i, \beta_i, \gamma_i)$ had already been fitted by education experts (Instituto Nacional de Estudos e Pesquisas Educacionais Anísio Teixeira, 2024). Therefore, for each one of the LLMs we considered, we only need to compute their ability parameters – given their response patterns. Intuitively, large values of $\theta$ correspond to test takers with high ability levels and vice versa. High ability value $\theta$ of an LLM implies better performance.

Although the ability levels of test takers can be used as a measure of their performance, one should also know if the test takers are *consistent* with the model, e.g., they should answer easy questions correctly if they answer difficult questions correctly. One index that enables us to evaluate the consistency of the test takers with the model is the $l_z$ index (De Ayala, 2013). Intuitively, the $l_z$ index is based on the standardization of a test-taker's log-likelihood function given their theta values. Assume a set of $I$ questions and test taker $j$ with ability $\theta_j$ and response vector $\mathbf{r}_j$ such that $\mathbf{r}_j(i) = 1$ (resp. $\mathbf{r}_j(i) = 0$) if $j$ answered question $i$ correctly (resp. wrongly). Then, the log-likelihood of $j$ is simply: $L_j = \sum_{i \in I} [\mathbf{r}_j(i) \ln P_{ij} + (1 - \mathbf{r}_j(i)) \ln(1 - P_{ij})]$. To standardize $L_j$ we need both its mean ($\mathbb{E}[L_j]$) and variance ($\text{Var}(L_j)$). Then, the $l_z$ *score* is computed as:

$$l_z(j) = \frac{L_j - \mathbb{E}[L_j]}{\sqrt{\text{Var}(L_j)}}. \tag{2}$$

In a well-designed test, the $l_z$ scores are expected to have a unit normal distribution – this is the case for humans taking the ENEM test (see for example Figure 3). In general, $l_z$ values close to 0 are considered good: it means the test takers' response patterns are consistent with what is expected from them by the model. Negative $l_z(j)$ scores reflect an unlikely response vector. A positive $l_z(j)$ score indicates that $j$ has a more likely response vector than indicated by their ability.

We can access the amount of information that an item $i$ provides to estimate $\theta$ under the 3PL model by the Fisher information, which is given by:

$$\mathcal{I}_i(\theta) = \alpha_i^2 \left[ \frac{(p_i - \gamma_i)^2}{(1 - \gamma_i)^2} \right] \left[ \frac{1 - p_i}{p_i} \right]. \tag{3}$$

The total information of a test is simply the sum of item information, i.e.,

$$\mathcal{I}(\theta) = \sum_{i \in I} \mathcal{I}_i(\theta).$$

The Fisher information is connected with the standard error of the estimation, given by $SE(\theta) = 1/\sqrt{\mathcal{I}(\theta)}$. When a test has high Fisher information in a certain $\theta$ range, the test has more discriminative power in that range, producing scores with fewer measurement errors.

## 4 METHODS

### 4.1 THE ENEM EXAM

The *Exame Nacional do Ensino Médio* (ENEM), world's second largest university entrance exam behind Chinese's Gaokao exam, is taken by millions of Brazilian students each year (Silveira & Mauá, 2018). ENEM comprises questions requiring different levels of domain-specific knowledge and reasoning (Almeida et al., 2023).

The exam is in Brazilian Portuguese and consists of four sections, each of which has 45 multiple-choice questions with five options (Instituto Nacional de Estudos e Pesquisas Educacionais Anísio Teixeira, 2024). Each section is treated as a separate exam for the purposes of modeling via IRT. The four sections consist of the **Humanities**, the **Languages and Codes**, the **Natural Sciences**, and the **Math** exams. The description of these exams is given in Appendix A.11.

Since 2009, the grades assigned to ENEM test-takers have been determined using IRT. Using IRT helps to penalize guessing, differentiate among students that otherwise would get the same (CTT) grade, and compare among students that took exams in different years. The ENEM organizers

release not only the exam content and questions, but also the student (anonymized) responses and their CTT and IRT scores, which enables downstream studies.

From our standpoint, there are a number of relevant aspects of the process used by the ENEM developers (Instituto Nacional de Estudos e Pesquisas Educacionais Anísio Teixeira, 2024). First, questions are given to a sample of students, whose answers are used to find inconsistencies and errors. Next, an important test of construct validity is to verify the unidimensionality of the latent trait, for which the ENEM team uses Full Information Factor Analysis (Bock et al., 1988). Finally, the IRT model itself is fit using the Marginal Maximum Likelihood Estimator (Bock & Aitkin, 1981). Using the results, the developers may exclude questions having poor model fit.

The exams, their solutions, and all the fitted parameters of the 3PL IRT model ($\theta_j, \alpha_i, \beta_i, \gamma_i$) are publicly available at the Brazilian government website (Instituto Nacional de Estudos e Pesquisas Educacionais Anísio Teixeira, 2024). To the best of our knowledge, these data are the largest and most comprehensive public dataset based on item-response theory available. The datasets contain questions and complete response patterns of all students taking the exams in 2022 and 2023. Questions for the 2023 exam were released in November 2023, minimizing the chance they are in training data for most of the LLMs we considered. However, we expect fragments of the exam being in the training data (e.g. poems, and any other widely available material used as part of a question) [1]. The number of test takers per year ranged from 2.2M to 3.7M.

The ENEM exams are initially made available as PDF files; we used the Python library *PyPDF2*, followed by regular expressions and some manual adjustments to extract each question from its exam file. In order to account for possible effects of Language, as diagnosed in previous work (Ranaldi & Pucci, 2023), we translated all questions to English and run all experiments in Portuguese and English. For those exam questions that incorporated images, we used the version of the exam designed for blind people containing textual descriptions of the images. We manually audited all questions in 2022 and 2023 exams to ensure their quality (Appendix A.1).

## 4.2 MODELS

We evaluate the following family of models: the open source models Mistral-7B, Gemma-7B, Llama2-7B, Llama2-13B, Llama3-8B, and GPT 3.5. For the open source models, we evaluate on both instructed and non-instructed tuned versions. Our choice of models enables the study of models of similar size (the majority of our models are of size 7B), but also introduces diversity of architectures (GPT, Gemma, Mistral, Llama), size (7B vs. 13B), training data (Llama2 vs. Llama3), and training strategies (with and without instruction tuning).

We prompt models with $\{0, 1, 4\}$-shots, following conventional question-answer benchmark prompting strategies (Robinson & Wingate, 2023) (example prompts in Appendix A.4). We measure model's next token probability across five option letters, and average predictions across 30 *shuffles* of the order of the answer choices to correct for the well-known effect of position bias (Pezeshkpour & Hruschka, 2023) (Details in Appendix A.4).

## 5 RESULTS

In this section, we present our main findings. All the results we show here are for the 2023 ENEM exams, with four-shot prompting. Results for the 2022 ENEM exam and for zero-shot and one-shot prompting and for open source instructed tuned models are shown in Appendices A.6 – A.9. The results we show in this section are strongly consistent with the results we get for the 2022 ENEM exam and for one-shot prompting.

### 5.1 ACCURACY VS. ABILITY LEVEL

We first investigate how humans compare to LLMs when IRT parameter $\theta$ is used instead of accuracy (the metric that is employed in most LLM benchmarking, e.g., (Bubeck et al., 2023; Touvron et al., 2023)). In Figure 1 we plot the CTT score (accuracy) vs IRT score ($\theta$) for 30 shuffles of answer options for each model. The light blue background points correspond to the humans who took the

---

[1] Gemma models are released in 2024 and we suspect contamination issues from analysis in Appendix A.3.

exam. Each of the closed curves in the figure corresponds to one LLM, and shows the central 90% of the LMM's distribution.

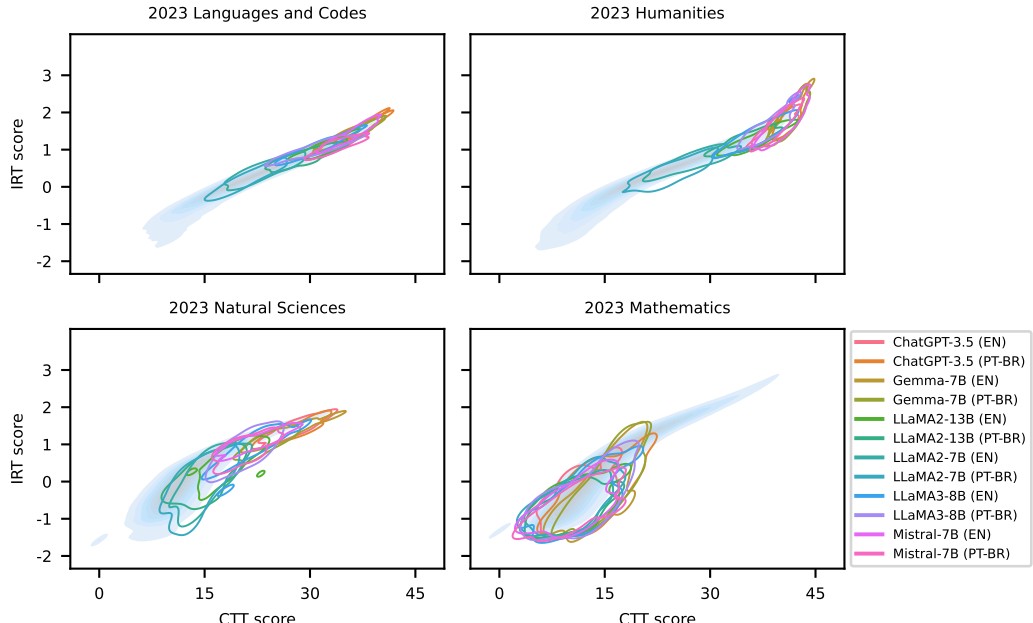

Figure 1: Distribution of CTT (accuracy) and IRT scores for humans and LLMs for the ENEM 2023 exam. LLMs are non-instructed tuned open source models and GPT3.5 with four-shot. LLM datapoints are computed from different shuffles of the order of answer choices.

First, we observe that there are many cases where identical accuracy scores result in different $\theta$ scores. This reflects the fact that IRT takes into account not just the number, but also the pattern of correct answers. Second, for many LLMs, particularly in the Humanities and Languages exams, there is overall greater variability in the accuracy score than in the IRT score. This suggests that IRT is less sensitive to the variations in LLM output that are due to the LLM's inherent randomness.

To compare the performance between LLMs and humans, we compare their IRT scores ($\theta$). Recall that IRT score of 0 corresponds to the average ability of a human test taker. Across all four subjects, the majority of models have CTT and IRT scores overlapping with humans. LLMs in general achieve $\theta$ scores above that of the human average in Humanities, Languages, and Natural Sciences, but below the human average in Mathematics. Looking at specific models, we find the Llama2 models at the lower end of $\theta$ scores, Mistral and Llama3 in the middle range, and GPT-3.5 and Gemma-7B at the higher end of $\theta$ scores.

The language of the exam affects some models' performance. In Languages and Natural Sciences, GPT-3.5 tends to perform better in Portuguese compared to English, while in Humanities and Natural Sciences, the Llama models tend to perform worse in Portuguese than in English. This suggests that there are differences regarding the reasoning ability and the amount of knowledge accessible to the models in each language.

Importantly, outlier models all tend to have higher accuracy and/or lower IRT scores than humans. These models answer more questions correctly than humans do, but show error patterns that are not entirely human-like. We dig into this phenomenon next.

## 5.2 RESPONSE PATTERNS

One of our goals is to assess whether the LLMs we examine show a good fit to the ENEM IRT model, as crafted by the educational expert team described in Section 4.1. Intuitively, a test taker showing a good fit to an IRT model is an individual $j$ that tends to make less frequent mistakes on "easy" questions (question $i$ with $\beta_i < \theta_j$) while making more frequent mistakes on "hard"

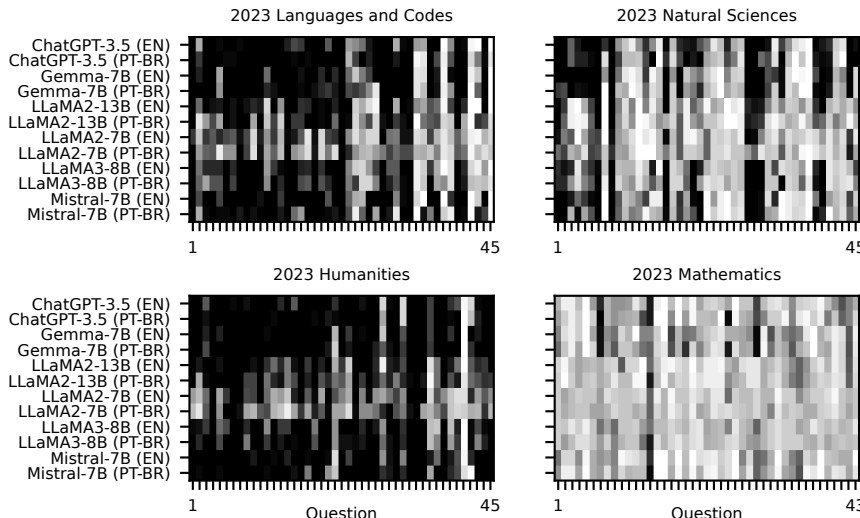

Figure 2: Response patterns for each LLM, where darker indicates more often correct (across random option shuffles). Questions are sorted in increasing difficulty ($\beta$ value). LLMs are non-instructed tuned open source models and GPT3.5 with four-shot.

questions (question $i$ with $\beta_i > \theta_j$). Thus, to assess fit we need to inspect the response patterns of the LLMs.

Figure 2 shows the response patterns of LLMs for the 2023 exam. Every cell $(i, j)$ corresponds to the probability that LLM $i$ answered question $j$ correctly, where probabilities are computed over the 30 shuffles. We use the gray scale with a black (resp. white) cell representing 1 (resp. 0). Questions are ordered in increasing order of their $\beta$ values. Generally, rows with darker overall patterns (higher correctness) are indicative of higher $\theta$ scores.

The figure demonstrates a number of points. For example, on the Math exam, the figure exhibits a response pattern that appears to show low $\theta$ values for all models, which confirms results in Figure 1. In addition, the figure shows that for some questions, the 30 shuffles of answer choices of a given model are often either all correct or all incorrect. However, there are some grey areas in the figure for all the exams, indicating that shuffling the options can affect the LLM's answers on certain items. Furthermore, the patterns show that many questions appear to be either "easy" (black) or "hard" (white) for all models at the same time. Likewise, in many cases, models show similar performance on the English and Portuguese versions of a given question.

Overall, the response patterns we observe suggest that the Math exam is "too difficult," with models often resorting to guessing. On the other hand, most LLMs consistently answer correctly the questions in the Humanities exam, implying that this is an easy exam for them. The performance of LLMs in the Natural Science exam is the most interesting as there are blocks of questions that most LLMs answer consistently correctly, interleaved with blocks of questions that most LLMs answer incorrectly. This suggests that there are questions that are easy for humans but difficult for LLMs and vice versa. In the next subsection, we analyze this phenomenon more closely.

## 5.3 RELIABILITY OF IRT SCORES FOR LLMS

In this section, we investigate whether the ENEM exam is a valid test for LLMs' ability, in the same way it is for humans. Intuitively, we want to define measures that allow us to quantify to what extent we trust the IRT scores we obtained for LLMs. We propose three different ways of doing this. The first is the *goodness-of-fit*, which quantifies whether the response of LLMs fit the IRT model. The second is based on *Fisher information*, measuring how much information the exam provides for estimating the $\theta$s in a certain range. Finally, we use the *discrimination index* which evaluates the capacity of questions to accurately distinguish between high and low performing test takers.

**Goodness-of-fit:** We use the $l_z$ score (see Section 3) to assess whether the test taker is behaving in a manner consistent with the model. Alternatively, we ask what is the *appropriateness* of a test-taker's estimated $\hat{\theta}$ as a measure of the test taker's true $\theta$? For example, imagine that an LLM has a response pattern of missing easy questions and correctly answering more difficult ones. Such a pattern may arise because the LLM was lucky on the hard questions, or it may arise because the LLM had access to memorized patterns that assisted in answering the hard questions. Generally, low $l_z$ scores suggest that the $\theta$ estimate of the model is less reliable (De Ayala, 2013).

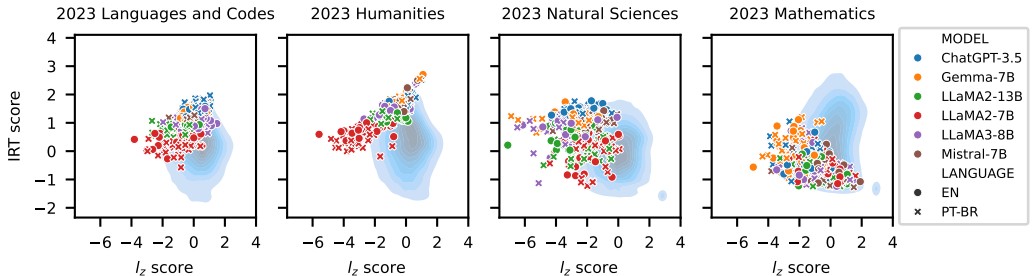

Figure 3: Distribution of $l_z$ and IRT scores for humans and LLMs. LLMs are non-instructed tuned open source models and GPT3.5 with 4-shot. LLM datapoints are computed from different shuffles.

In Figure 3 we show $l_z$ scores plotted against $\theta$ scores of LLMs across the four exams in 2023 (2022 is shown in Appendix A.9). As in previous plots, the light blue points in the background show the distribution of the same two scores for the human test takers. Starting again with the Math exam, we note that $l_z$ values are low, but now we can see that the response patterns of the LLMs are indeed quite human-like; LLMs behave like humans with similarly low $l_z$ values. One possible reason for this behavior is that the Mathematics exam tends to be the harder exam of ENEM, leading to more guessing, which may make the human $l_z$ values for Mathematics smaller.

For the Languages exam, models perform better in general (higher $\theta$ values) and the most $l_z$ scores being close to 0 (and with a similar spread as the human distribution of $l_z$'s) suggest that these $\theta$ estimates are reliable – the models are showing human-like response patterns.

The results become more nuanced as we look at the Natural Sciences exam. For this exam, most models, including the high-performing ones (i.e., GPT-3.5 and Gemma-7B), show values well outside the human distribution, with a long tail in the negative values of $l_z$. Comparing the GPT-3.5 and Gemma-7B results in Figures 1 and 4, we can infer that the high accuracy (CTT scores) achieved by these models on the Natural Sciences exam are quite misleading; although GPT-3.5 and Gemma-7B answer many questions correctly, their response pattern is very unlikely, with very low $l_z$ values. This corroborates with Figure 2, which shows an interchange of blocks of correct and incorrect answers from the models, creating an unlikely response pattern.

In Humanities, almost all LLMs perform reasonably well, achieving $\theta$ scores above zero (the average human level). However, Llama2-7B, while obtaining above average accuracy scores (Figure 1) and good $\theta$ scores, has low average $l_z$ scores. This suggests that the IRT scores for Llama2-7B may not be reliable. Examination of the corresponding rows in Figure 2 shows that this is the only model that does not have a consistent response pattern across shuffles, leading to the observed low $l_z$ score.

**Fisher Information:** We investigate further whether the ENEM exams are giving us accurate estimates of the LLMs ability levels from another standpoint – that of Fisher Information (see Section 3, Equation equation 3). Intuitively, Fisher Information quantifies whether there was enough information in the test to infer the ability level of a test taker at a certain ability level. Figure 4 shows, for every ENEM exam, the total Fisher Information $\mathcal{I}(\theta)$ on the top plot, and the $\theta$ scores for the models (95% Confidence Interval (CI) computed using the shuffles) on the bottom plot. This plot reinforces the observation that for some models in Natural Sciences and for all models in Mathematics, the models' $\theta$ are not in the range of the exam with the highest information – the models' ability levels fall in the tail of the Fisher Information histogram. Hence, *the Math exam is not useful for making meaningful measurements of these LLMs,* casting doubt on the informativeness of the models' $\theta$ scores on this exam. The lack of discrimination ability of this exam is reflected by the responses

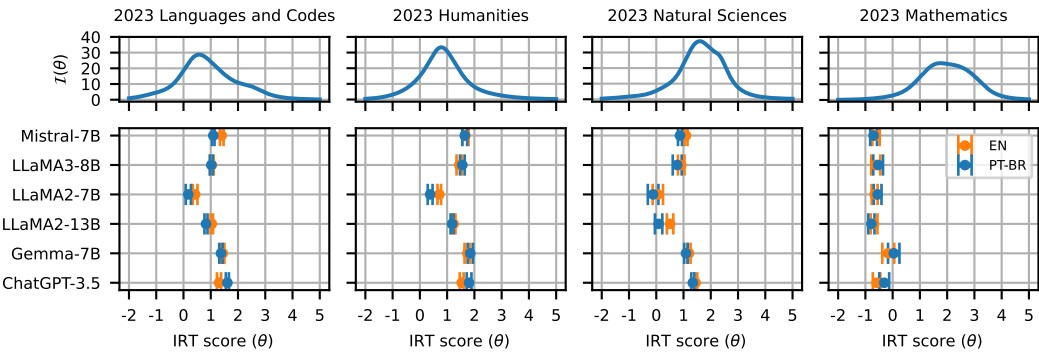

Figure 4: Total Fisher information of the exams and the IRT scores (95% Confidence Interval (CI)) for LLMs. LLM datapoints are computed from different shuffles.

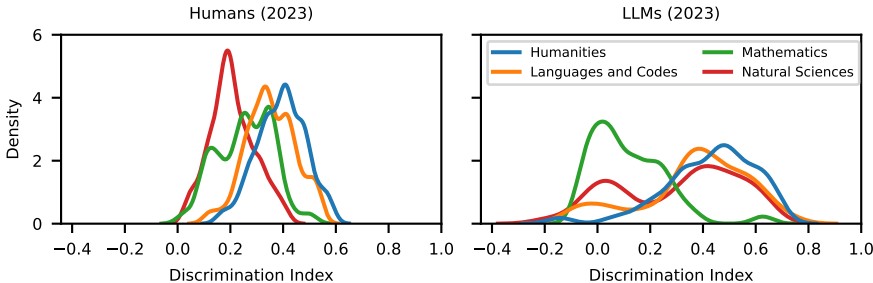

Figure 5: Discrimination Indices for questions in the 2023 exam for both Humans and LLMs.

for many models showing apparently random response patterns in the corresponding heatmap (see Figure 2).

**Discrimination Index:** To further assess the reliability of the IRT scores, we also turn into psychometrics and use the notion of the item *discrimination index (DI)*, which measures how well an item on a test distinguishes between high and low scorers on the entire test (Chan, 2015). Let $P_h$ (resp. $P_l$) be the proportion of the top 25% (resp. low 25%) LLMs (in terms of $\theta$, including the shuffles) that correctly answer the item; then $DI = P_h - P_l$, the difference of the two proportions. DI ranges from -1 to 1, and questions with DI higher than 0.2 are considered good, while lower DI indicates flaws (Wu & Adams, 2007).

Figure 5 shows the distribution of the discrimination indices computed for humans and LLMs for the 2023 exam. Overall, we notice that discrimination indices computed for LLMs are more negative compared to those of humans. We also observe that a significant fraction of Math questions have low discriminative power, reinforcing the hypothesis that this exam is not well designed to measure Math abilities for LLMs. Nonetheless, the Humanities and Languages have several questions with very good discriminative power. Interestingly, the Natural Sciences exam appears to follow a bimodal distribution, containing both informative and poorly-designed questions. This may reflect the fact that the Natural Sciences exam is a hybrid test, containing a mix of knowledge-based items and items that demand more complex reasoning over numbers and images, which can be less useful for evaluating the current state-of-the-art LLMs.

**Attributes affecting the reliability of IRT scores:** In a further investigation, shown in Appendix A.2, we explore potential causes of low discrimination. We investigate item attributes such as the existence of images or numbers in the questions as we believe that these attributes impede LLMs from understanding the question properly. Our preliminary results suggest that LLMs' ability to understand math questions and parse images is sub-par compared to their capacity to answer pure text-based questions. In Appendix A.10 we show examples of non-discriminating and highly

discriminating items for the 2023 Natural Sciences exam. In Appendix A.3, we reach a similar conclusion by looking at model accuracy against model perplexity, a model intrinsic metric.

# 6 CONCLUSIONS

The ongoing debate in LLM evaluation centers around whether exams designed for humans are appropriate tools for measuring the performance of LLMs. In this paper, we provide a case study that illustrates methods that can be used to address this question, as well as specific results for a range of current LLMs. We leverage the largest known human exam for which a public IRT model is available, and show that IRT can be leveraged to distinguish between human-like and non-human-like responses under the model. We show cases where LLMs respond in non-human-like ways and show how to identify those cases using a model-fit metric. Further, we show that using IRT we can determine when an exam is capable of making meaningful measurement of an LLM's ability in a given subject area. Using our evaluation framework, we find that the ENEM Math exam is not appropriate to make meaningful measurements of the models' ability, for the LLMs we study. At the same time, Humanities and Language exams are better suited for evaluating the LLMs' abilities on those subjects. We conclude that IRT modeling, drawing on a long history of psychometric theory, provides a set of crucial tools for assessing whether exams designed for humans are actually meaningful measures of LLM ability. Our results suggest that they should be used in future studies when questions are raised regarding the performance of LLMs on human exams.

# 7 REPRODUCIBILITY STATEMENT

We have made the following efforts to ensure the reproducibility of our research:

- We provide the detailed prompts used in the research in the Appendix;
- We include a supplementary material file containing all input files and code used to produce the plots and tables in the paper;
- Upon acceptance, we will make available a GitHub page containing all the code needed to reproduce the results in the paper.

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

| language | subject | Accuracy (CTT) | $\theta$ | $l_z$ |
|---|---|---|---|---|
| en | humanities | $29.5 \pm 10.7$ | $-0.57 \pm 0.56$ | $-1.25 \pm 1.18$ |
| en | languages | $24.7 \pm 8.6$ | $-0.99 \pm 0.49$ | $-0.39 \pm 1.03$ |
| en | science | $25.5 \pm 8.2$ | $-0.34 \pm 0.53$ | $-0.74 \pm 1.25$ |
| en | math | $22 \pm 6.3$ | $-0.6 \pm 0.34$ | $-0.66 \pm 0.97$ |
| pt-br | humanities | $24 \pm 7$ | $-0.83 \pm 0.38$ | $-0.83 \pm 1.08$ |
| pt-br | languages | $23.1 \pm 7.1$ | $-1.06 \pm 0.42$ | $-0.32 \pm 1.01$ |
| pt-br | science | $23.5 \pm 7.3$ | $-0.48 \pm 0.41$ | $-0.5 \pm 1.18$ |
| pt-br | math | $23.2 \pm 6.4$ | $-0.55 \pm 0.4$ | $-0.86 \pm 1.05$ |

Table 1: Random choice selection performance on English and Portuguese versions of 2022 test 4 subjects.

## A    SUPPLEMENTAL MATERIAL

### A.1    MANUAL AUDITING OF EXAM QUESTIONS

Assuming the original questions written by the ENEM authorities are good test instruments for testing student capability, we focus on ensuring the quality of adapted dataset for LLM evaluation. We manually correct the artifacts for each question in 2022 and 2023. In the next sections, we describe the artifacts from those easier to address (sec A.1.2 A.1.3), to deeper-rooted problems (i.e., harder to correct, sec A.1.4), as well as how we addressed them manually (sec A.1.5).

### A.1.1    LABEL ACCURACY

We assume answers are correct as translation and parsing of single characters can be quite reliable, and that the original ENEM test is tested across millions of human test takers and will be discarded if it had a wrong answer. When we look at the label distribution for 2022, options "ABCDE" each occur 39/39/37/36/33 times, making it fairly balanced. We also ran random baselines on the same option shuffles as the model (Table 1).

### A.1.2    TRANSLATION ARTIFACTS

We found several issues pertaining to initial round of translation in this dataset. Mainly, independent translation of question context and answer option leads to incoherence. Details are sometimes mis-translated ("p.d.d" translated to "d.d.p"). There are many non-standardized translations pertaining to chemical formulas, proper nouns, and mathematical formulas. In general, there are significant amount of awkward phrasing, incomplete translation, and linguistic idiosyncrasies lost in translation.

**Independent translation of context and question**    In a few cases, the answer options are expected to complete the last sentence of the question. After translation, options do not all fit as completions of the sentence (Q11). Translation without context also leads to improper translation of polysemantic terms. "Coagulation" maybe translated correctly in the question, but becomes "coagulating" as a stand-alone word (Q96). "Good" and "fair" (when used as survey options) gets translated to "regular" and "I will" as stand-alone options (Q171)

**Inconsistent translation details**    Within the same questions, there are cases where the same concept is translated differently. In one question, the context introduces the concept "potential difference (p.d.d)", and later referred to it as "d.d.p" and "d.p.d". Within different options, the same unit can sometimes be plural and sometimes be singular (when it should be consistently plural)

**Non-standard translation**    1) Chemical formula translation is non-standard. "$N2O3$" becomes "$N$ $2O3$", and "$NH4+$" becomes "$NH4$ positively charged". 2) (Proper) nouns are sometimes capitalized when they shouldn't. For instance, one question begins with the sentence "*On the Gravitational Field of a Mass Point According to Einstein's Theory A 'Black Hole is a...*" 3) Mathematical equations are overly verbatim. This we suspect is partially due to an issue with using audio version of the

test. For example, if an option is the formula $9(\frac{8!}{(8-2)!2!} - 1)$, its Portuguese representation would be "*9 vezes ( ( (8 fatorial dividido por ( (8 menos 2) fatorial vezes 2 fatorial)) menos 1)*" and the English translation exacerbates the situation by translating parenthesis literally as well: "*9 times open parenthesis, open parenthesis, 8 factorial divided by, open parenthesis, open parenthesis, 8 minus 2, close parenthesis, factorial times 2 factorial, close parenthesis, close parenthesis, minus 1, close parenthesis.*". Sometimes, delimiters are omitted after translation: "9,300" becomes "9 300".

**Awkward phrasings** There exist awkward phrasings throughout translation. They range from causing minor difficulty in understanding (i.e., "*Life: the science of biology Bears, because they are not truly hibernating, wake up due to the presence of thermogenin, a mitochondrial protein that prevents protons from reaching ATP synthase, generating heat.*") to sometime completely non-sense (i.e., "*articulation of several narrative nuclei*")

**Incomplete translation** There is no fine line between proper code switching (where proper nouns should remain in Portuguese script) to in-complete translation. The amount of Portuguese left over range from single words, to phrases in options (not consistently across options), to entire sentences within the question.

**Linguistic idiosyncrasies lost in translation** In one question, the problem arises when English translation does not match with literal tokens of expressions in Portuguese ("*Next to the man is the message: "Men don't cry", with a large X drawn over the word "no"*"). The word "no" does not appear in the English phrase "Men don't cry" but the statement as a whole makes sense in the Portuguese version of the instruction. In a separate question, the topic is on testing for a Portuguese specific pronoun inflection. However, when it was translated into one single word in English, the question no longer makes sense ("*They told me... - They told me. - Huh? - The correct word is "they told me". Not "they told me". - I speak the way I want to. And I'll tell you more... Or is it "tell you"? - What's that? - I'm telling you that you... - "You" and "you" don't go together...*")

### A.1.3 DOCUMENT PARSING ARTIFACTS

Each section consistently contains an error of this kind, where the last part of the question got wrongfully parsed into part of the first option (option (A)). In a separate instance, a figure was wrongfully parsed into one of the options of the previous question. In the Portuguese version of the exam, structural components of the question (e.g., title, subtitle, caption) are consistently concatenated together without proper separation. This often leads to incoherent English translations.

### A.1.4 AUDIO-VERSION ARTIFACTS

Audio description of images, tables, and figures are not always sufficient, or the most intuitive. For instance, a question asks test taker to note why a particular painting stands out, and the answer is due to the painting's "distortion when representing human figure", which is difficult to qualitatively describe, no matter how complete the description of an image is. Similarly, textual description of geometric figures can be impossibly complicated ("*...Figure of a grid with 7 horizontal and 7 vertical lines, on which a polygonal path is drawn by means of a continuous line on the grid lines, joining the starting point P , located on the second vertical line, from left to right, and between the sixth and seventh horizontal lines, from top to bottom, to the end point Q , which is located between the sixth and seventh vertical lines, from left to right, and on the second horizontal line, from top to bottom...*")

### A.1.5 MANUAL CORRECTION

The majority of the artifacts begin with incorrect parsing of the PDF documents related to structural components. To address this, we manually audited each question, and added correct spacing and newlines to each question. These improvements result in better translations from DeepL API qualitatively. After translation, we make minimal edits to improve syntactic and semantic issues

through Grammarly to obtain a score of at least 95 [2] [3]. For each answer option, we ensure consistent part-of-speech, especially if they are sentence completions of the questions. For math and science sections, we follow consistent markdown-like format the same way as other mathematical reasoning datasets Hendrycks et al. (2020); Cobbe et al. (2021); Yu et al. (2023). Here we list the full set of modification rules for 2022 (question numbers are referenced in parenthesis):

- Separate description of the image by '\n' before and after.
- "Por cento" becomes %.
- Number in the form 7 000 becomes 7000.
- From "abre aspas" "fecha aspas" to """.
- Remove "Descrição da estrutura química", "Descrição do esquema", "Descrição da associação de baterias", "Descrição da imagem" from the options".
- "De carga positiva" to +, "De carga negativa" to -, "de carga dois menos" to (2-).
- For a subset of the questions, we follow the non-blind version of the question (157, 158, 163, 166, 168, 171, 174, 177, 178, 179)
- Remove period at the end options or questions of math questions (to avoid confusion).

Here are the list of rules we use for English version of the exam (2022):

- Change number decimal from "3,1415" to "3.1415".
- Manual translation fix (49, 162).

### A.1.6 LIMITATIONS OF THE DATASET

There are a few limitations of the dataset:

1. Even though the English version of the exam is modified manually, there are still issues with the presentation of the questions. We rely mostly on Grammarly feedback, but it is not perfect. Our judgement of how fluently a question is written is also subjective. The ideal method would be to recruit professional human translators, which is costly and time consuming.

2. The content of many of the questions are focused on knowledge common to Brazilian culture, or problems in Brazilian society. The English translations may not cover the full extent of cultural, language specific phenomenons or connotations.

3. We assume the transcription of images and tables to be sufficient for the models to understand and solve the question.

### A.2 ATTRIBUTES THAT AFFECT GOODNESS-OF-FIT

Given that questions have wide range of discrimination indices for LLMs, we investigate a potential cause described in the psychometrics literature for aberrant response patterns: lack of *subabilities* Meijer (1996), i.e., specific skills required to answer a question correctly. We hypothesize that some item attributes, such as whether the question contains images or numbers in its statement or among the options, may be disproportionately harder for LLMs and hence represent subabilities that explain the aberrant response patterns quantified in Figure 3.

We built a contingency table relating non-discriminative/discriminative items (i.e., items with discriminative index lower/higher than 0.2) and the aforementioned attributes, and run a $\chi^2$ independence test. The results for the Natural Sciences exam are shown in Table 2. For this exam, we observe high $\chi^2$ values which indicate that the abilities of the LLM models with respect to math reasoning and interpreting images are sub-par compared to their capacity in solving pure text questions. While Language and Humans exams are most purely text and the Math exam mostly demands

---

[2]grammarly.com/

[3]We chose not to use a large model such as GPT3.5 to rephrase the translations because it may artificially lower the perplexity and change the meaning of the questions.

Table 2: $\chi^2$ test for the correlation between poorly-discriminating items and item attributes in the Natural Sciences exam in 2022 and 2023. Significant values are in bold. High values of $\chi^2$ indicate that images or numbers make the item less useful to evaluate the LLMs we experiment with.

| Item Attribute | 2022 | 2023 |
|---|---|---|
| Contains images | 0.401 (0.052) | **3.906** (0.048) |
| Contains numbers in the answers | **7.331** (0.007) | **6.264** (0.012) |
| Contains numbers in the statement | **3.961** (0.046) | 3.212 (0.073) |

reasoning with images and numbers, the nature of the Natural Sciences exam is hybrid, containing both types of questions. This may well explain the bimodal distribution of discrimination indices in Figure 5 and the aberrant response patterns identified by the very low $l_z$ scores in Figure 3, and highlights how psychometrics can aid the design of better and more valid benchmarks for LLMs.

### A.3 MODEL ACCURACY RELATION TO MODEL PERPLEXITY

One reason that models may error differently than humans is due to their training corpus. If models have encountered similar question or topics, if not identical, to those in our dataset during training, they may perform unexpectedly well, even if the questions are difficult. Recent work in data contamination proposed a few model intrinsic metrics that can be used to detect contamination Shi et al. (2023). Mainly, the Min-k% Prob score takes the average probability of the top-k percentile tokens with minimum probabilities [4]:

$$\text{MIN-K\% Prob (x)} = -\frac{1}{E} \sum_{x_i \in \text{Min-K\%(x)}} \log p(x_i | x_1, \ldots, x_{i-1}) \tag{4}$$

where $x = x_1, x_2, \ldots, x_N$ denotes the input sequence of N tokens, Min-K% Prob(x) represents the set containing tokens with minimum k percentile probabilities, and $E$ represents the size of such set. Note here that Min-k% Prob is intrinsic to each model, and if a model has been exposed to more similar training data as the questions, its Min-k% Prob would be low for that question.

We do not expect any model to have unexpectedly low Min-K% Prob(x) on any of our questions, considering it is highly unlikely that the ENEM questions were parsed and translated to English, and somehow ended up in the training corpus. What we are more interested here, is whether such score is correlated to model's accuracy on the answer predictions. If they are negatively correlated (i.e. high Min-K% Prob corresponds to low accuracy), this is evidence for the hypothesis that training on related data leads to higher accuracy.

To investigate this hypothesis, we plot 4-shot model accuracy (averaged across 31 option shuffles) against Min-20% Prob for four subjects in exam 2022 in English along with the Pearson correlations [5] in Figure 6. In all except 1 model-subject pair (Llama2 chat in humanities, we investigate this further) do we see a significant negative correlation ($p < 0.05$) between accuracy and Min-k 20% Prob, indicate that model doesn't necessarily do better if they have encountered similar data during training. Another way to interpret this, is that it is not likely that these models have seen our data during training.

**The few negative correlation cases** As seen before, we observe a significant negative correlation for Llama-2 7B Chat in humanities. To get a full understanding of whether this is a stand-alone phenomenon, we examine Portuguese version of the exam, as well as exam in 2023, and show our findings below in Table 3. We do not see the same correlation in the Portuguese version of the exam. However, we additionally see Gemma-it negatively correlated with humanities section in both English and Portuguese version of the exam in 2023, as well as Gemma with languages section in 2023. The later two correlations are robust across a few other metrics we investigated from Shi et al. (2023) as well, we think this may suggest data contamination, but we cannot test such hypothesis because Gemma training data is not public.

---

[4]We follow the equation in https://github.com/swj0419/detect-pretrain-code/blob/main/src/run.py
[5]https://docs.scipy.org/doc/scipy/reference/generated/scipy.stats.pearsonr.html

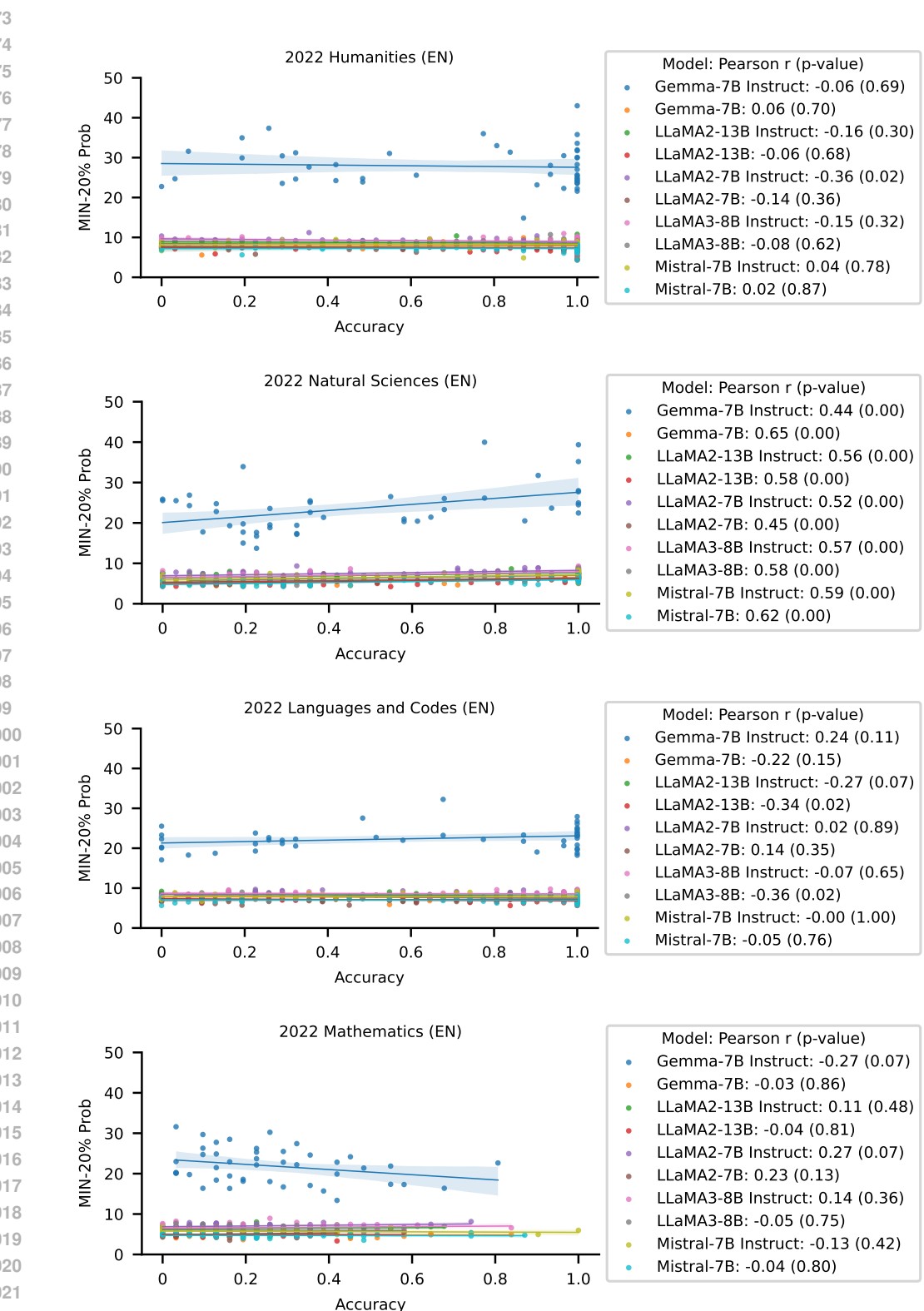

Figure 6: Model Min-20% Prob vs. 4-shot accuracy across four subjects in 2022 in English

**Positive correlations in 2022 science**  In 2022 Science, both English and Portuguese, we see significant *positive* correlation across all models (Table 3).

Through qualitative analysis, we find that the questions with highest perplexities were formatted more in a sentence completion-like structure similar to Question 1. Whereas less perplexity questions involve more image/table description with reasoning needed to obtain the answer (question 2). This is similar to what we discover with discriminative index in Section **??** in the main text.

Listing 1: high perplexity question with high model accuracy.

```
Question: Technique modifies rattlesnake venom protein to create a
    drug that modulates blood clotting

Rattlesnake venom can cause life-threatening hemorrhaging to those
    bitten by the snake. However, researchers from Brazil and
    Belgium have developed a molecule of pharmaceutical interest,
    PEG-collinein-1, from a protein found in the snake's venom.
    The molecule is capable of modulating blood clotting. Although
     the technique is not new, it was applied for the first time
    from an animal toxin in its recombinant form, i.e. produced in
     the laboratory by a genetically modified fungus.

This new drug has potential applications for
Options:
(A) prevent the formation of thrombi, typical in some cases of
    stroke.
(B) treat the consequences of profound anemia, due to the loss of
    a large volume of blood.
(C) prevent the manifestation of urticaria, commonly related to
    allergic processes.
(D) reduce swelling of the lymph nodes, part of the immune
    response to different infections.
(E) regulate the fluctuations in blood pressure characteristic of
    hypertension.
```

Listing 2: low perplexity question with low model accuracy.

```
Question: On a hot day, two colleagues are playing with the water
    from the hose. One of them wants to know how high the water
    jet reaches from the outlet when the hose is positioned
    vertically. The other colleague then proposes the following
    experiment: they position the water outlet of the hose in a
    horizontal direction, 1 meter above the ground, and then
    measure the horizontal distance between the hose and the place
     where the water hits the ground. The measurement of this
    distance was 3 meters, and from this, they calculated the
    vertical reach of the water jet. Consider the acceleration of
    gravity to be 10 meters per second squared.

The result they obtained was
Options:
(A) 1.50 meter.
(B) 2.25 meters.
(C) 4.00 meters.
(D) 4.50 meters.
(E) 5.00 meters.
```

We also tried filtering for top N percent most difficult questions per subject and recalculate all the correlations. We did not find any significant difference to results above.

| year | lang | subj | L2-7b | L2-7b-it | L2-13b | L2-13b-it | L3-8b-it | L3-8b | M-7b | M-7b-it | G-7b-it | G-7b |
|---|---|---|---|---|---|---|---|---|---|---|---|---|
| 2022 | en | CH | -0.14/0.36 | **-0.36/0.02** | -0.06/0.68 | -0.16/0.30 | -0.15/0.32 | -0.08/0.62 | 0.02/0.87 | 0.04/0.78 | -0.06/0.69 | 0.06/0.70 |
| | | LC | 0.14/0.35 | 0.02/0.89 | **-0.34/0.02** | -0.27/0.07 | -0.07/0.65 | **-0.36/0.02** | -0.05/0.76 | -0.00/1.00 | 0.24/0.11 | -0.22/0.15 |
| | | CN | **0.45/0.00** | **0.52/0.00** | **0.58/0.00** | **0.56/0.00** | **0.57/0.00** | **0.58/0.00** | **0.62/0.00** | **0.59/0.00** | **0.44/0.00** | **0.65/0.00** |
| | | MT | 0.23/0.13 | 0.27/0.07 | -0.04/0.81 | 0.11/0.48 | 0.14/0.36 | -0.05/0.75 | -0.04/0.80 | -0.13/0.42 | -0.27/0.07 | -0.03/0.86 |
| | pt | CH | -0.09/0.56 | -0.12/0.43 | -0.06/0.70 | -0.05/0.73 | -0.07/0.65 | -0.05/0.74 | -0.09/0.56 | -0.06/0.69 | -0.20/0.18 | 0.18/0.24 |
| | | LC | 0.10/0.53 | -0.02/0.88 | -0.06/0.67 | -0.05/0.73 | 0.08/0.61 | -0.20/0.20 | 0.14/0.35 | -0.09/0.56 | 0.14/0.37 | -0.21/0.16 |
| | | CN | **0.41/0.01** | **0.42/0.00** | **0.49/0.00** | **0.48/0.00** | **0.57/0.00** | **0.52/0.00** | **0.53/0.00** | **0.52/0.00** | **0.46/0.00** | **0.58/0.00** |
| | | MT | -0.17/0.26 | -0.15/0.34 | 0.12/0.44 | -0.02/0.91 | 0.07/0.66 | -0.08/0.59 | -0.18/0.23 | -0.14/0.35 | -0.05/0.76 | 0.12/0.42 |
| 2023 | en | CH | -0.06/0.72 | -0.07/0.66 | -0.09/0.56 | 0.06/0.69 | -0.20/0.20 | -0.18/0.23 | -0.20/0.18 | -0.07/0.65 | **-0.32/0.03** | -0.16/0.30 |
| | | LC | -0.06/0.67 | -0.22/0.15 | **-0.31/0.04** | -0.24/0.12 | -0.21/0.17 | **-0.30/0.04** | -0.18/0.23 | -0.08/0.61 | -0.05/0.76 | **-0.32/0.03** |
| | | CN | 0.21/0.17 | 0.21/0.17 | **0.31/0.04** | 0.16/0.31 | **0.30/0.05** | 0.28/0.06 | 0.14/0.35 | 0.15/0.34 | 0.20/0.19 | 0.24/0.11 |
| | | MT | 0.17/0.28 | -0.07/0.66 | -0.04/0.82 | -0.02/0.87 | -0.05/0.75 | 0.15/0.35 | 0.03/0.85 | 0.19/0.21 | 0.06/0.68 | 0.16/0.32 |
| | pt | CH | -0.00/1.00 | -0.02/0.92 | 0.09/0.58 | 0.18/0.25 | -0.02/0.90 | -0.11/0.46 | -0.04/0.77 | 0.01/0.96 | **-0.30/0.05** | -0.09/0.55 |
| | | LC | -0.21/0.16 | -0.23/0.13 | -0.27/0.07 | -0.17/0.26 | -0.20/0.18 | -0.24/0.11 | -0.18/0.23 | -0.10/0.53 | -0.13/0.40 | **-0.36/0.02** |
| | | CN | 0.11/0.49 | 0.17/0.26 | 0.25/0.10 | 0.04/0.82 | 0.14/0.37 | **0.36/0.01** | 0.15/0.32 | 0.14/0.35 | 0.08/0.61 | 0.13/0.41 |
| | | MT | -0.01/0.96 | 0.02/0.87 | -0.02/0.90 | -0.04/0.79 | -0.07/0.67 | 0.06/0.71 | -0.08/0.60 | 0.09/0.56 | 0.18/0.24 | 0.28/0.06 |

Table 3: Correlation between model accuracy and Min-k% Prob across exam, languages, and subjects for all models (**L2**=llama2, **L3**=Llama3, **M**=Mistral, **G**=gemma, **it**=instruction-tuned/chat). The first number indicates the coefficient of the correlation, and the second, the p-value. Entries with p-value $< 0.05$ are in **bold**. **CN**=Humanities, **LC**=Languages, **CN**=Sciences, **MT**=Math
.

## A.4 PROMPTING DETAILS

To administering the test to LLMs, we measure the next token logits across the 5 letter options directly (i.e. letter "A", "B", "C", "D", "E"), and take the argmax as the model's choice (invariant to sampling temperature). We shuffle the option orders (30 runs) and take the average to calibrate model's prior on generating each letter options. For API-based model (GPT3.5), we query for 1 token generation, and obtain top-20 logits, and use that for our prediction. In the sections below we include 0-shot (Listing 3), 1-shot (Listing 4, 5, 6, 7), and 4-shot prompts (Listing 8) we use in main experiments. For 1-shot, we choose the 1-shot example for each of the four subjects by selecting the easiest question (i.e., with lowest $\beta$) from the same subject in the 2021 exam. For 4-shot, we concatenate the 1-shots from four subjects and shuffle the options to evenly distribute the answer among five option letters.

**Potential limitations** We ran exploratory experiments with Chain-of-Thought (CoT) like prompting (Wei et al., 2022), but and did not see significant changes. We did not include the results because CoT prompting requires generating reasoning strings and parsing answers, making 30-shuffles extremely slow to run for all models. Future directions could explore how much effect more complex prompting techniques have in assimilating model behaviors. Regarding the best prompting strategy, we do acknowledge recent criticisms on first letter evaluation(Wang et al., 2024). At the time of our writing, it is still the best evaluation strategy for multiple choice question-answering data. We also acknowledge that there are more capable models than GPT3.5 that is available through API services but as our work is not trying to identify the SOTA model we did not feel the need to evaluate latest and largest models. Lastly, we assume Portuguese and Brazilian culture is present in the training data for the language models we test. Future work could evaluate the amount of multilingual training's affect on some of these IRT metric we propose.

Listing 3: 0-shot prompt used across all four subjects.

```
Here are some questions from a college entrance exam. Choose the
    correct answer to the best of your ability, and output in the
    following format:
Answer: (Option)

Question: {QUESTION}
Options:
(A) {OPTION_A}
(B) {OPTION_B}
(C) {OPTION_C}
(D) {OPTION_D}
(E) {OPTION_E}
```

```
Answer: (
```

Listing 4: 1-shot prompt used for Natural Science.

```
Here are some questions from a college entrance exam. \\ Choose
    the correct answer to the best of your ability, and output in
    the following format:
Answer: (Option)

Question:
Buffalos are animals considered rustic by breeders and are
    therefore left in the field without reproductive control.
    Because of this type of breeding, inbreeding is common,
    leading to the appearance of diseases such as albinism and
    heart defects, among others. Separating the animals properly
    by sex would minimize the occurrence of these problems.

What prior biotechnological procedure is recommended in this
    situation?

Options:
(A) Transgenics.
(B) Gene therapy.
(C) DNA vaccine.
(D) Genetic mapping.
(E) Therapeutic cloning.

Answer: (D) Genetic mapping.

Question: {QUESTION}
Options:
(A) {OPTION_A}
(B) {OPTION_B}
(C) {OPTION_C}
(D) {OPTION_D}
(E) {OPTION_E}
Answer: (
```

Listing 5: 1-shot prompt used for Math.

```
Here are some questions from a college entrance exam. Choose the
    correct answer to the best of your ability, and output in the
    following format:
Answer: (Option)

Question:
A hamburger chain has three franchises in different cities. To
    include a new type of snack on the menu, the chain's marketing
     manager suggested putting five new types of snacks on sale in
     special editions. The snacks were offered for the same period
     of time to all the franchisees. The type with the highest
    average sold per franchise would be permanently included on
    the menu. At the end of the trial period, management received
    a report describing the quantities sold, in units, of each of
    the five types of snacks in the three franchises.

Image description: The table shows the quantity sold of each type
    of snack in franchises 1, 2, and 3.
```

```
Franchise 1 sold 415 type-1 snacks, 395 type-2 snacks, 425 type-3
    snacks, 430 type-4 snacks, and 435 type-5 snacks.
Franchise 2 sold 415 type-1 snacks; 445 type-2 snacks; 370 type-3
    snacks; 370 type-4 snacks and 425 type-5 snacks.
Franchise 3 sold 415 type-1 snacks; 390 type-2 snacks; 425 type-3
    snacks; 433 type-4 snacks and 420 type-5 snacks.

Based on this information, the management has decided to include
    the following type of snack on the menu

Options:
(A) 1
(B) 2
(C) 3
(D) 4
(E) 5

Answer: (E) 5

Question: {QUESTION}
Options:
(A) {OPTION_A}
(B) {OPTION_B}
(C) {OPTION_C}
(D) {OPTION_D}
(E) {OPTION_E}
Answer: (
```

Listing 6: 1-shot prompt used for Humanities.

```
Here are some questions from a college entrance exam. Choose the
    correct answer to the best of your ability, and output in the
    following format:
Answer: (Option)

Question:
The situation of the working class in England
Friedrich Engels

At the same time, thanks to the ample opportunities I have had to
    observe the middle classes, your adversaries, I have quickly
    concluded that you are right, absolutely right, not to expect
    any help from them. Its interests are diametrically opposed to
     yours, even if it constantly tries to claim the opposite and
    wants to persuade you that it feels the greatest sympathy for
    your lot. But her actions belie her words.

In the text, the author presents ethical outlines that correspond
    to

Options:
(A) the foundation of the idea of surplus value.
(B) concept of class struggle.
(C) fundamentals of the scientific method.
(D) paradigms of the inquiry process.
(E) domains of commodity fetishism.

Answer: (B) concept of class struggle.
```

```
Question: {QUESTION}
Options:
(A) {OPTION_A}
(B) {OPTION_B}
(C) {OPTION_C}
(D) {OPTION_D}
(E) {OPTION_E}
Answer: (
```

Listing 7: 1-shot prompt used for Languages.

```
Here are some questions from a college entrance exam. Choose the
    correct answer to the best of your ability, and output in the
    following format:
Answer: (Option)

Question:
Sinha
Chico Buarque and Joao Bosco

If the owner bathed
I wasn't there
By God our Lord
I didn't look Sinha
I was in the fields
I'm not one to look at anyone
I'm not greedy anymore
I can't see straight

Why put me in the trunk
Why hurt me
I swear to you
I've never seen Sinha
[...]
Why carve up my body
I didn't look at Sinha
Why would you
You'll pierce my eyes
I cry in Yoruba
But I pray for Jesus
So that you can
Take away my light

In this fragment of the song's lyrics, the vocabulary used and the
     situation portrayed are relevant to the country's linguistic
    heritage and identity, in that

Options:
(A) physical and symbolic violence against enslaved people.
(B) value the influences of African culture on national music.
(C) relativize the syncretism that makes up Brazilian religious
    practices.
(D) narrate the misfortunes of the love relationship between
    members of different social classes.
(E) problematize the different worldviews in society during the
    colonial period.

Answer: (A) physical and symbolic violence against enslaved people
```

```
Question: {QUESTION}
Options:
(A) {OPTION_A}
(B) {OPTION_B}
(C) {OPTION_C}
(D) {OPTION_D}
(E) {OPTION_E}
Answer: (
```

Listing 8: 4-shot prompt used across all four subjects.

```
Here are some questions from a college entrance exam. Choose the
    correct answer to the best of your ability, and output in the
    following format:
Answer: (Option)

Question:
Buffalos are animals considered rustic by breeders and are
    therefore left in the field without reproductive control.
    Because of this type of breeding, inbreeding is common,
    leading to the appearance of diseases such as albinism and
    heart defects, among others. Separating the animals properly
    by sex would minimize the occurrence of these problems.

What prior biotechnological procedure is recommended in this
    situation?

Options:
(A) Transgenics.
(B) Gene therapy.
(C) DNA vaccine.
(D) Genetic mapping.
(E) Therapeutic cloning.

Answer: (D) Genetic mapping.

Question:
Sinha
Chico Buarque and Joao Bosco

If the owner bathed
I wasn't there
By God our Lord
I didn't look Sinha
I was in the fields
I'm not one to look at anyone
I'm not greedy anymore
I can't see straight

Why put me in the trunk
Why hurt me
I swear to you
I've never seen Sinha
[...]
Why carve up my body
I didn't look at Sinha
Why would you
You'll pierce my eyes
I cry in Yoruba
```

```
But I pray for Jesus
So that you can
Take away my light

In this fragment of the song's lyrics, the vocabulary used and the
    situation portrayed are relevant to the country's linguistic
    heritage and identity, in that

Options:
(A) physical and symbolic violence against enslaved people.
(B) value the influences of African culture on national music.
(C) relativize the syncretism that makes up Brazilian religious
    practices.
(D) narrate the misfortunes of the love relationship between
    members of different social classes.
(E) problematize the different worldviews in society during the
    colonial period.

Answer: (A) physical and symbolic violence against enslaved people

Question:
The situation of the working class in England
Friedrich Engels

At the same time, thanks to the ample opportunities I have had to
    observe the middle classes, your adversaries, I have quickly
    concluded that you are right, absolutely right, not to expect
    any help from them. Its interests are diametrically opposed to
     yours, even if it constantly tries to claim the opposite and
    wants to persuade you that it feels the greatest sympathy for
    your lot. But her actions belie her words.

In the text, the author presents ethical outlines that correspond
    to

Options:
(A) the foundation of the idea of surplus value.
(B) concept of class struggle.
(C) fundamentals of the scientific method.
(D) paradigms of the inquiry process.
(E) domains of commodity fetishism.

Answer: (B) concept of class struggle.

Question:
A hamburger chain has three franchises in different cities. To
    include a new type of snack on the menu, the chain's marketing
     manager suggested putting five new types of snacks on sale in
     special editions. The snacks were offered for the same period
     of time to all the franchisees. The type with the highest
    average sold per franchise would be permanently included on
    the menu. At the end of the trial period, management received
    a report describing the quantities sold, in units, of each of
    the five types of snacks in the three franchises.

Image description: The table shows the quantity sold of each type
    of snack in franchises 1, 2, and 3.
Franchise 1 sold 415 type-1 snacks, 395 type-2 snacks, 425 type-3
    snacks, 430 type-4 snacks, and 435 type-5 snacks.
```

```
Franchise 2 sold 415 type-1 snacks; 445 type-2 snacks; 370 type-3
    snacks; 370 type-4 snacks and 425 type-5 snacks.
Franchise 3 sold 415 type-1 snacks; 390 type-2 snacks; 425 type-3
    snacks; 433 type-4 snacks and 420 type-5 snacks.

Based on this information, the management has decided to include
    the following type of snack on the menu

Options:
(A) 1
(B) 2
(C) 3
(D) 4
(E) 5

Answer: (E) 5

Question: {QUESTION}
Options:
(A) {OPTION_A}
(B) {OPTION_B}
(C) {OPTION_C}
(D) {OPTION_D}
(E) {OPTION_E}
Answer: (
```

## A.5 COMPUTE RESOURCES

We used GPUs (V100 or A100) provided by a university cluster[6]. For the main experiments, we used around 200 hours of GPU time (roughly 20 hours per model). Moreover, we used the OpenAI API to run the experiments with GPT3.5.

## A.6 ZERO AND ONE SHOT PROMPTING RESULTS FOR 2023

### A.6.1 CTT AND IRT $\theta$

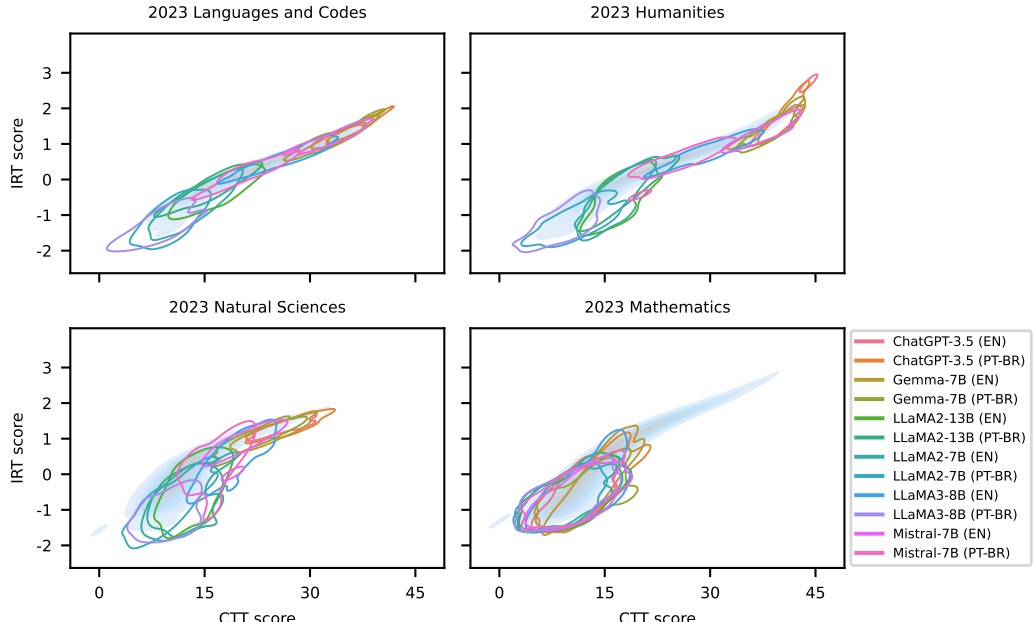

Figure 7: Distribution of CTT (accuracy) and IRT scores for humans and LLMs for the ENEM 2023 exam. LLMs are non-instructed tuned open source models and GPT3.5 with zero-shot. LLM datapoints are computed from different shuffles.

---

[6]We will disclose it after the reviewing phase due to the double-blind process.

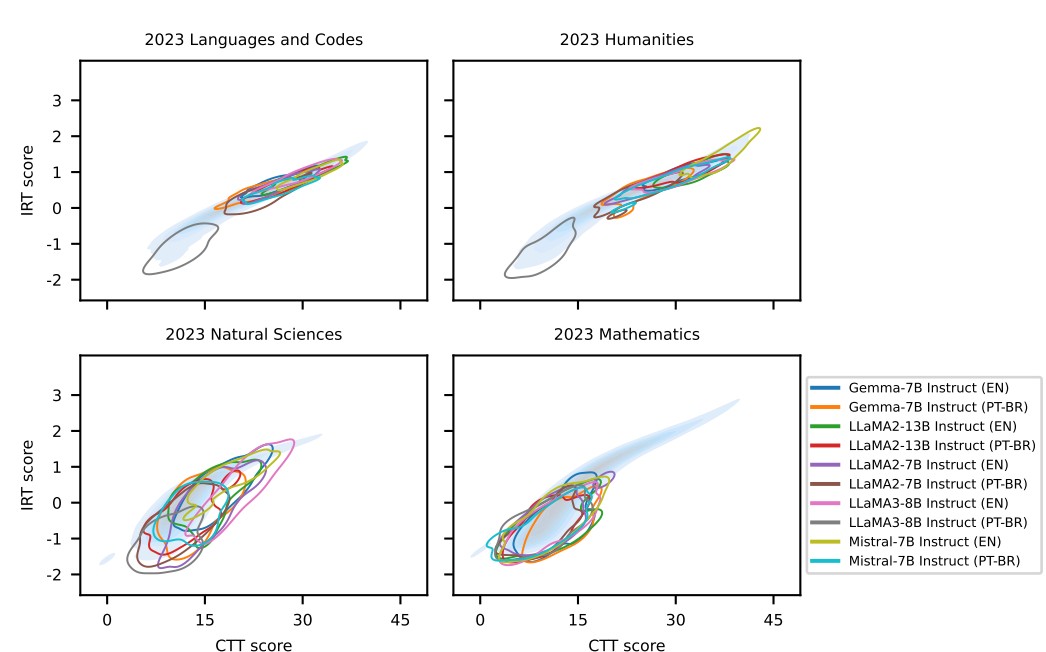

Figure 8: Distribution of CTT (accuracy) and IRT scores for humans and LLMs for the ENEM 2023 exam. LLMs are instructed tuned open source models with zero-shot. LLM datapoints are computed from different shuffles.

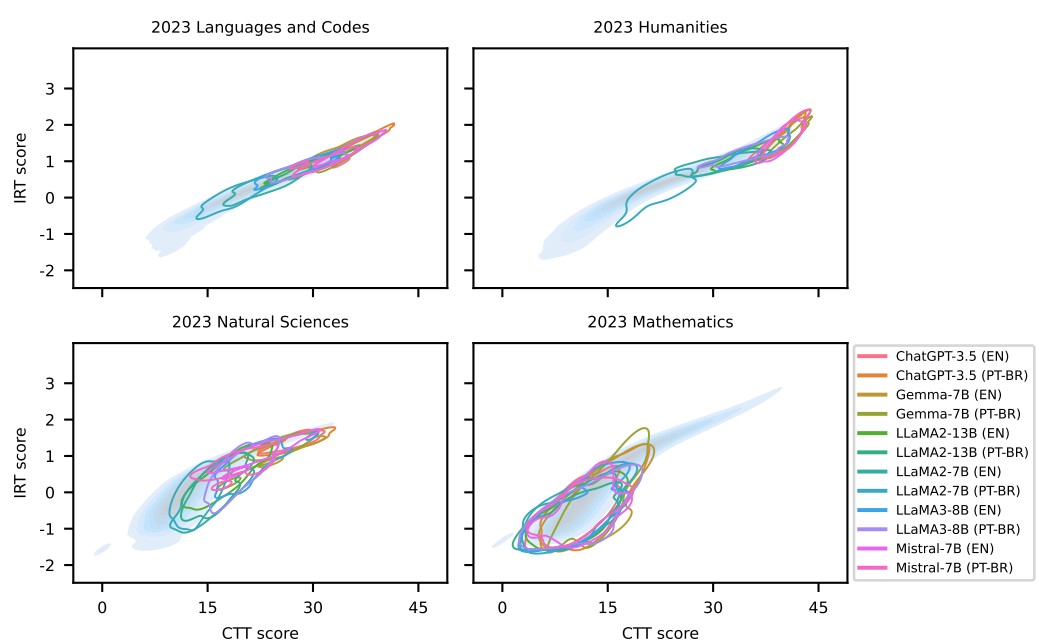

Figure 9: Distribution of CTT (accuracy) and IRT scores for humans and LLMs for the ENEM 2023 exam. LLMs are non-instructed tuned open source models and GPT3.5 with one-shot. LLM datapoints are computed from different shuffles.

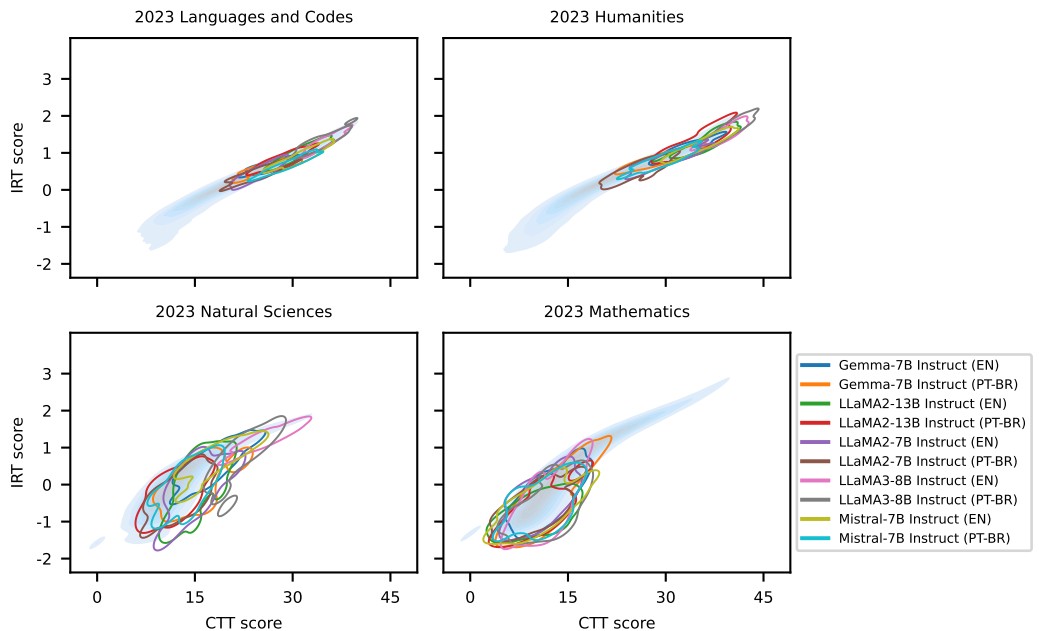

Figure 10: Distribution of CTT (accuracy) and IRT scores for humans and LLMs for the ENEM 2023 exam. LLMs are instructed tuned open source models with one-shot. LLM datapoints are computed from different shuffles.

### A.6.2 RESPONSE PATTERNS

We show 43 items for the 2023 Math exam, instead of 45, because 2 items failed to converge and produce item parameters when the ENEM organizers fitted the human model.

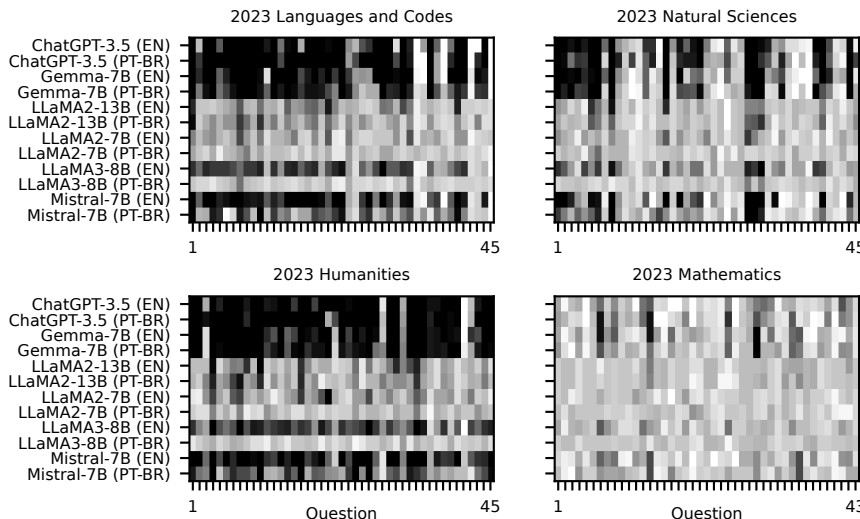

Figure 11: Response patterns for each LLM, where darker indicates more often correct. Questions are sorted by difficulty ($\beta$ value). LLMs are non-instructed tuned open source models and GPT3.5 with zero-shot.

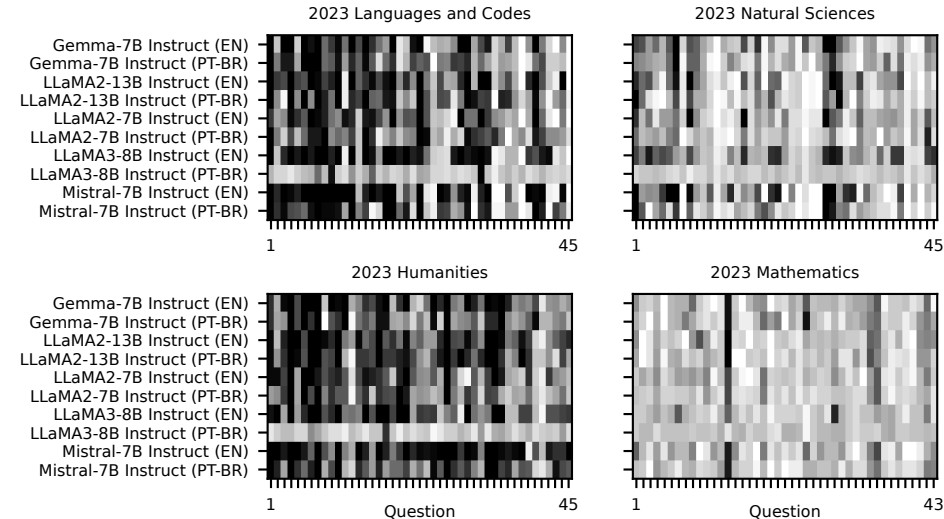

Figure 12: Response patterns for each LLM, where darker indicates more often correct. Questions are sorted by difficulty ($\beta$ value). LLMs are instructed tuned open source models with zero-shot.

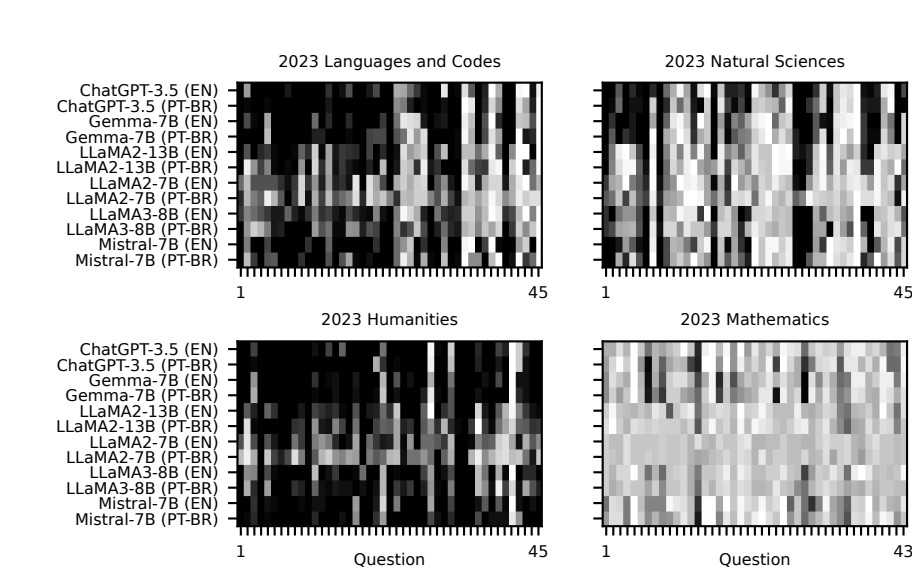

Figure 13: Response patterns for each LLM, where darker indicates more often correct. Questions are sorted by difficulty ($\beta$ value). LLMs are non-instructed tuned open source models and GPT3.5 with one-shot.

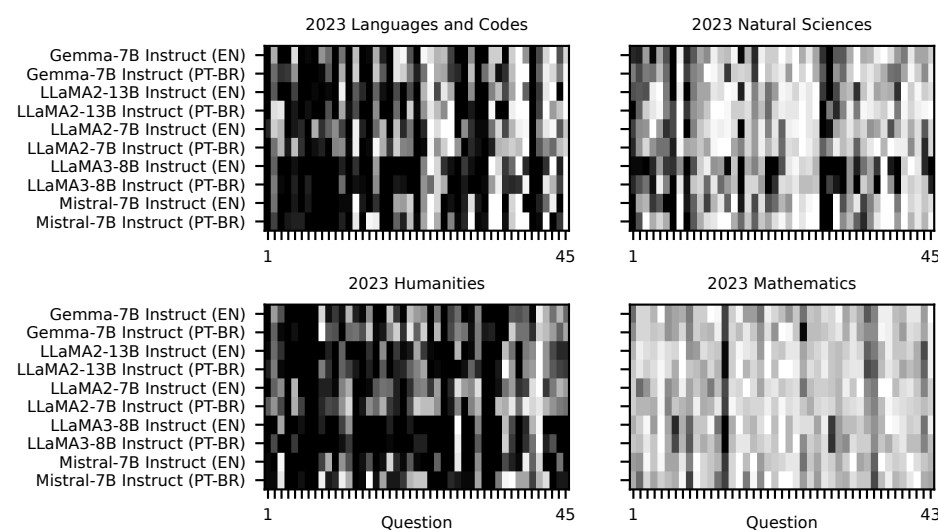

Figure 14: Response patterns for each LLM, where darker indicates more often correct. Questions are sorted by difficulty ($\beta$ value). LLMs are instructed tuned open source models with one-shot.

### A.6.3 COMPARING IRT $\theta$ AND $l_z$

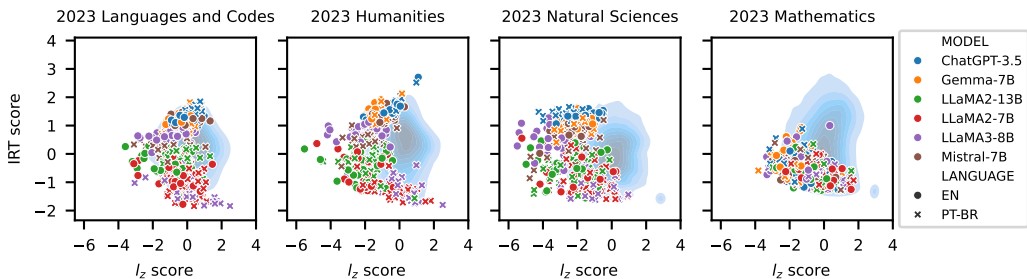

Figure 15: Distribution of $l_z$ and IRT scores for humans and LLMs in the ENEM 2023 exam. LLMs are non-instructed tuned open source models and GPT3.5 with zero-shot. LLM datapoints are computed from different shuffles.

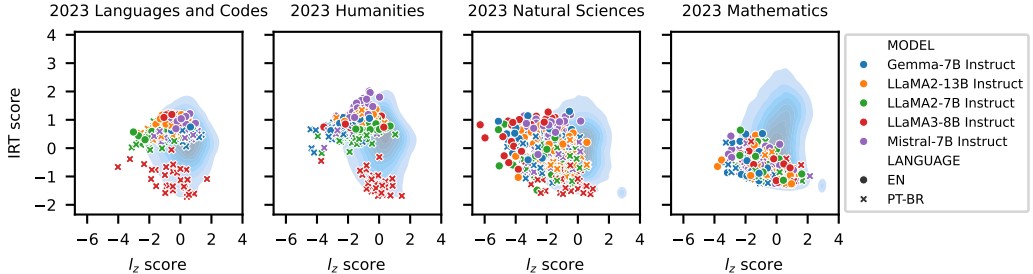

Figure 16: Distribution of $l_z$ and IRT scores for humans and LLMs in the ENEM 2023 exam. LLMs are instructed tuned open source models with zero-shot. LLM datapoints are computed from different shuffles.

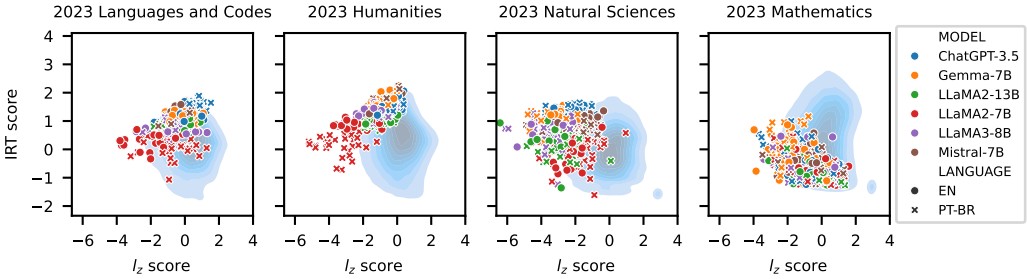

Figure 17: Distribution of $l_z$ and IRT scores for humans and LLMs in the ENEM 2023 exam. LLMs are non-instructed tuned open source models and GPT3.5 with one-shot. LLM datapoints are computed from different shuffles.

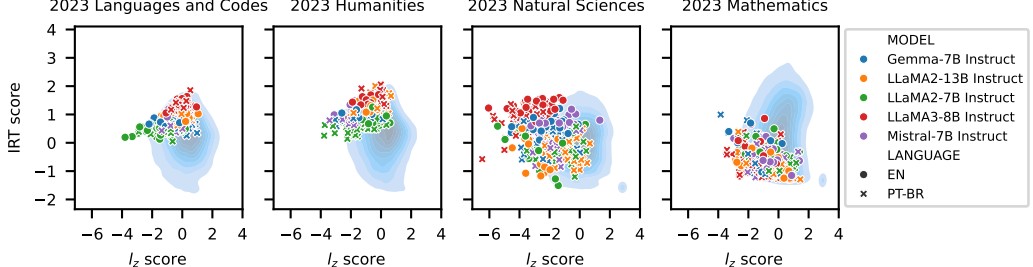

Figure 18: Distribution of $l_z$ and IRT scores for humans and LLMs in the ENEM 2023 exam. LLMs are instructed tuned open source models with one-shot. LLM datapoints are computed from different shuffles.

## A.7 CTT AND IRT $\theta$ FOR 2022

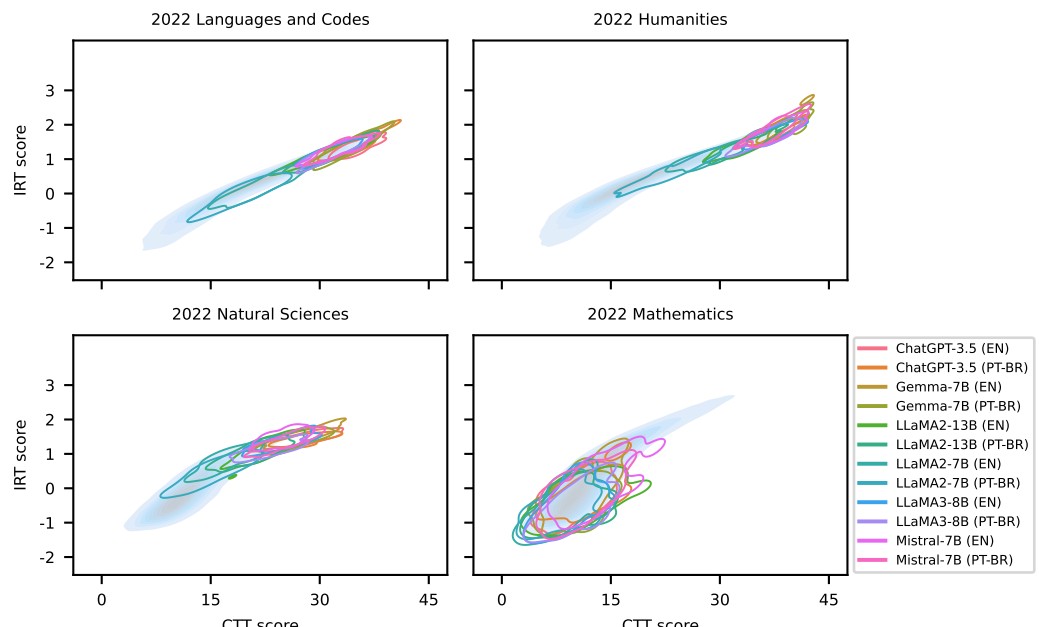

Figure 19: Distribution of CTT (accuracy) and IRT scores for humans and LLMs for the ENEM 2022 exam. LLMs are non-instructed tuned open source models and GPT3.5 with four-shot. LLM datapoints are computed from different shuffles.

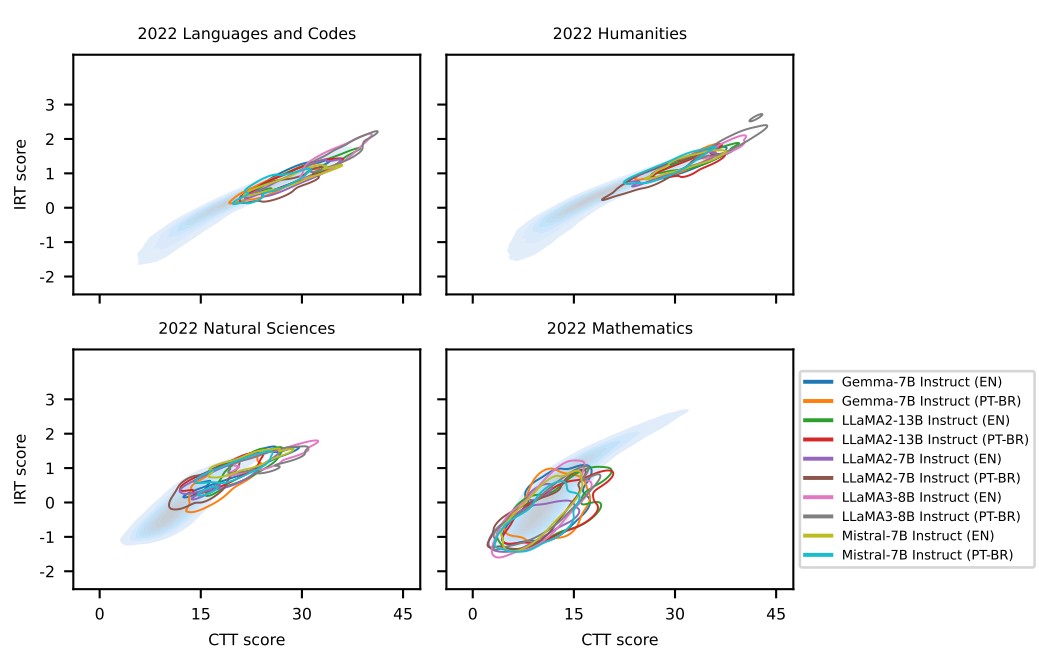

Figure 20: Distribution of CTT (accuracy) and IRT scores for humans and LLMs for the ENEM 2022 exam. LLMs are instructed tuned open source models with four-shot. LLM datapoints are computed from different shuffles.

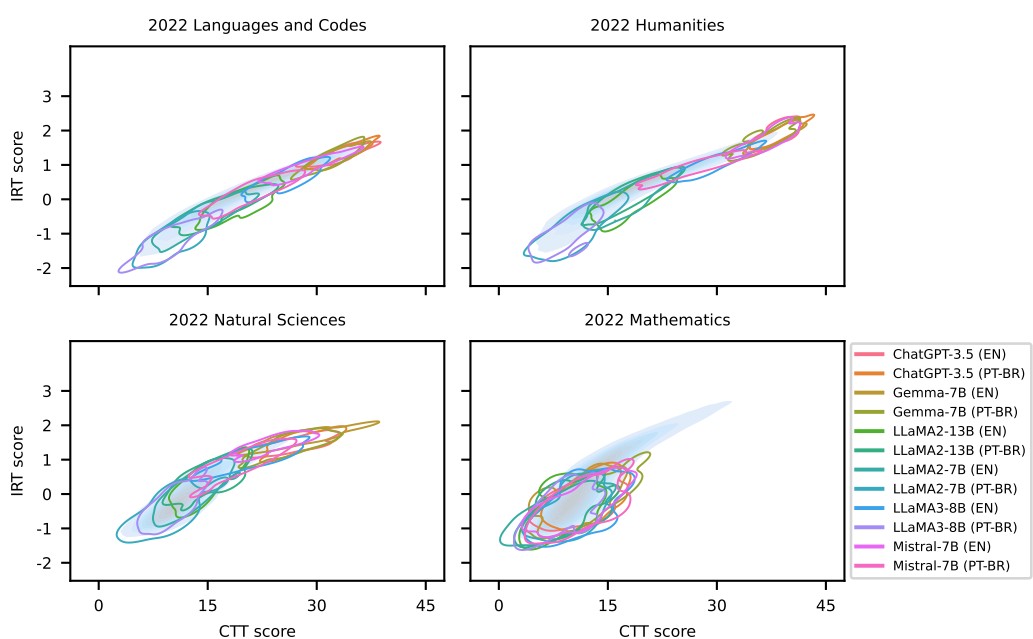

Figure 21: Distribution of CTT (accuracy) and IRT scores for humans and LLMs for the ENEM 2022 exam. LLMs are non-instructed tuned open source models and GPT3.5 with zero-shot. LLM datapoints are computed from different shuffles.

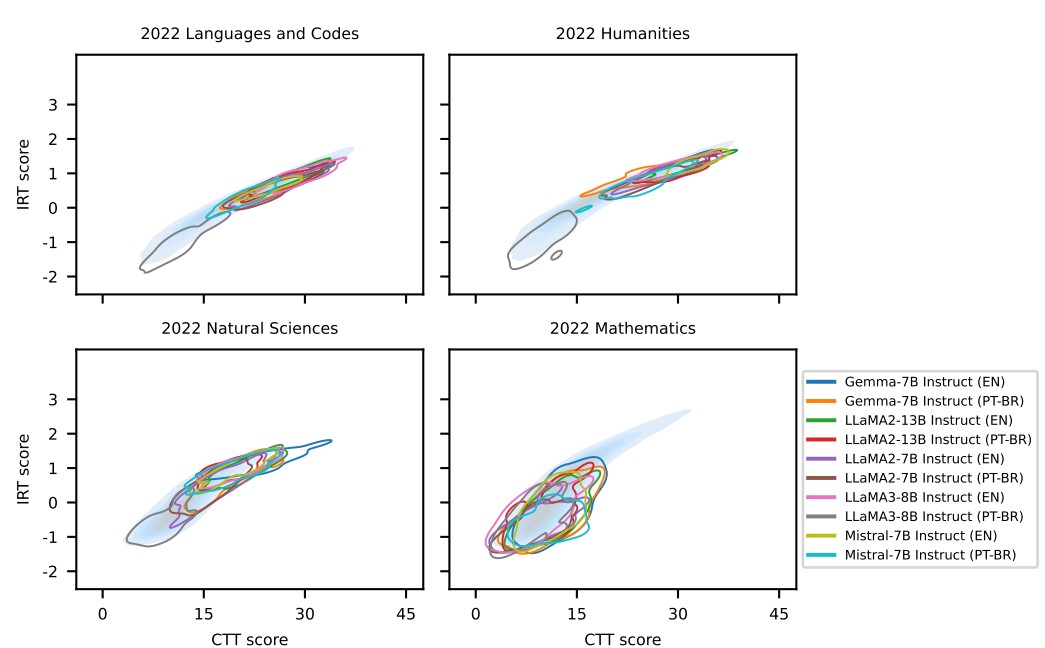

Figure 22: Distribution of CTT (accuracy) and IRT scores for humans and LLMs for the ENEM 2022 exam. LLMs are instructed tuned open source models with zero-shot. LLM datapoints are computed from different shuffles.

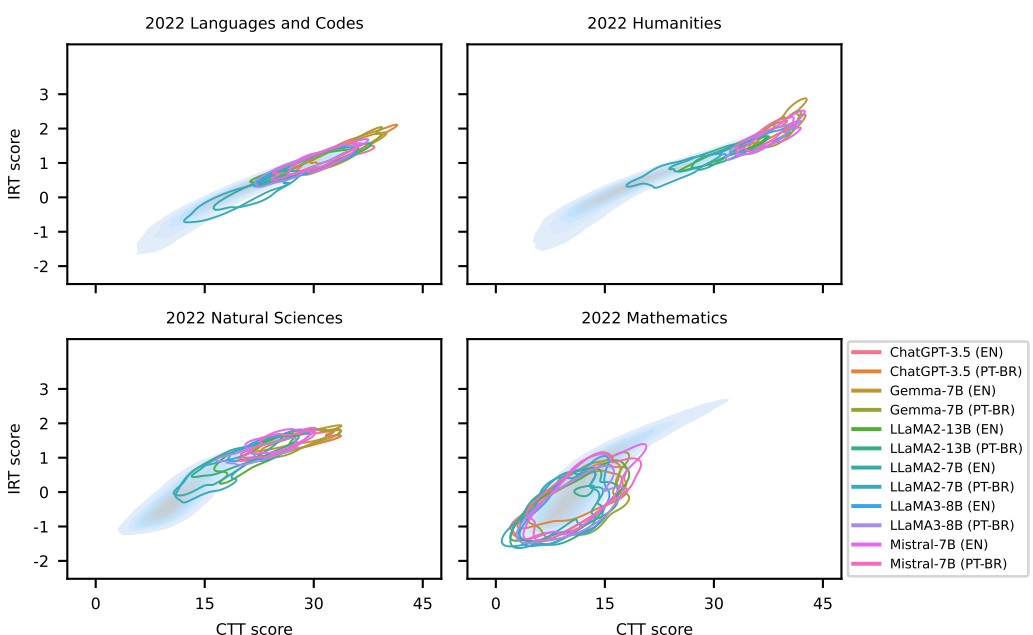

Figure 23: Distribution of CTT (accuracy) and IRT scores for humans and LLMs for the ENEM 2022 exam. LLMs are non-instructed tuned open source models and GPT3.5 with one-shot. LLM datapoints are computed from different shuffles.

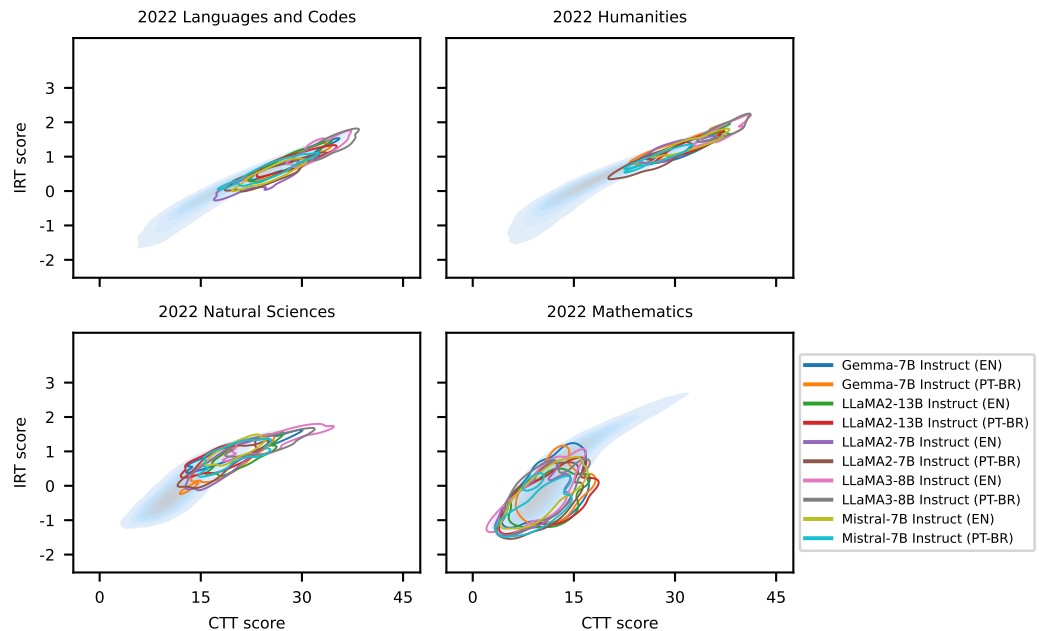

Figure 24: Distribution of CTT (accuracy) and IRT scores for humans and LLMs for the ENEM 2022 exam. LLMs are instructed tuned open source models with one-shot. LLM datapoints are computed from different shuffles.

## A.8 RESPONSE PATTERNS FOR 2022

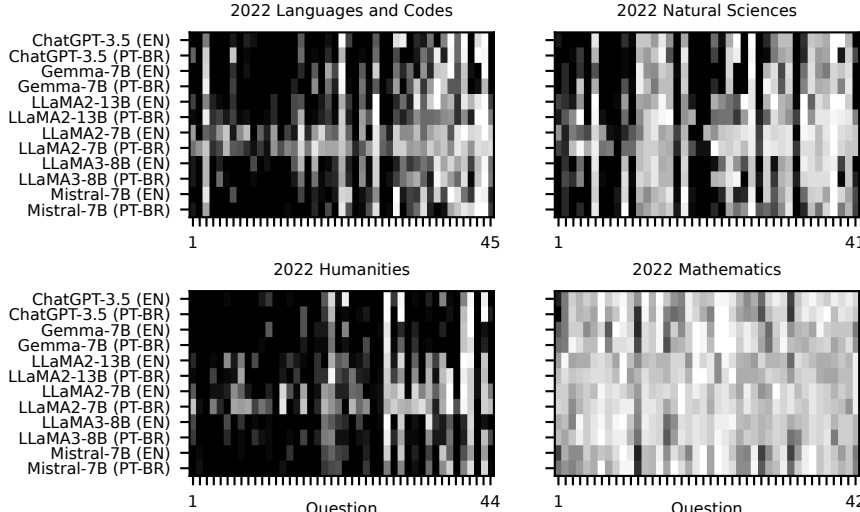

Figure 25: Response patterns for each LLM, where darker indicates more often correct. Questions are sorted by difficulty ($\beta$ value). LLMs are non-instructed tuned open source models and GPT3.5 with four-shot.

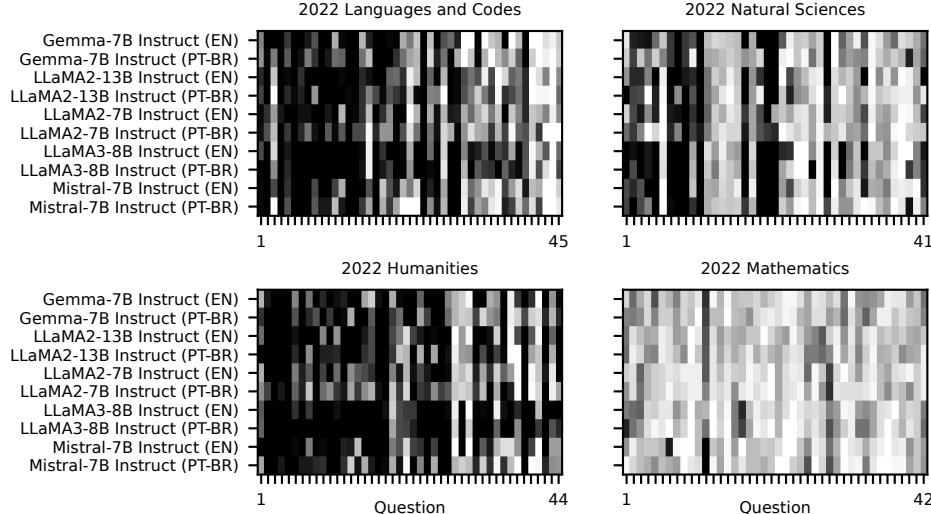

Figure 26: Response patterns for each LLM, where darker indicates more often correct. Questions are sorted by difficulty ($\beta$ value). LLMs are instructed tuned open source models with four-shot.

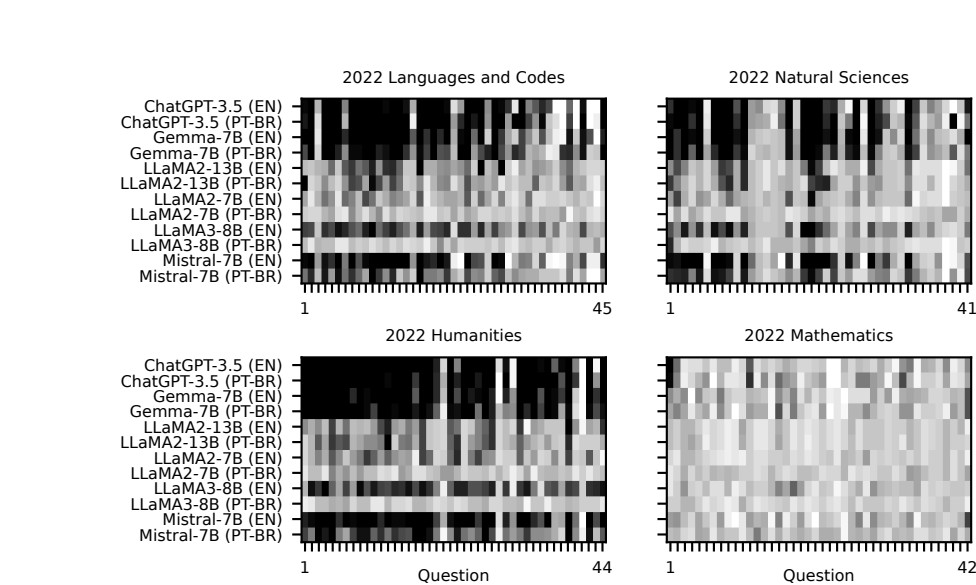

Figure 27: Response patterns for each LLM, where darker indicates more often correct. Questions are sorted by difficulty ($\beta$ value). LLMs are non-instructed tuned open source models and GPT3.5 with zero-shot.

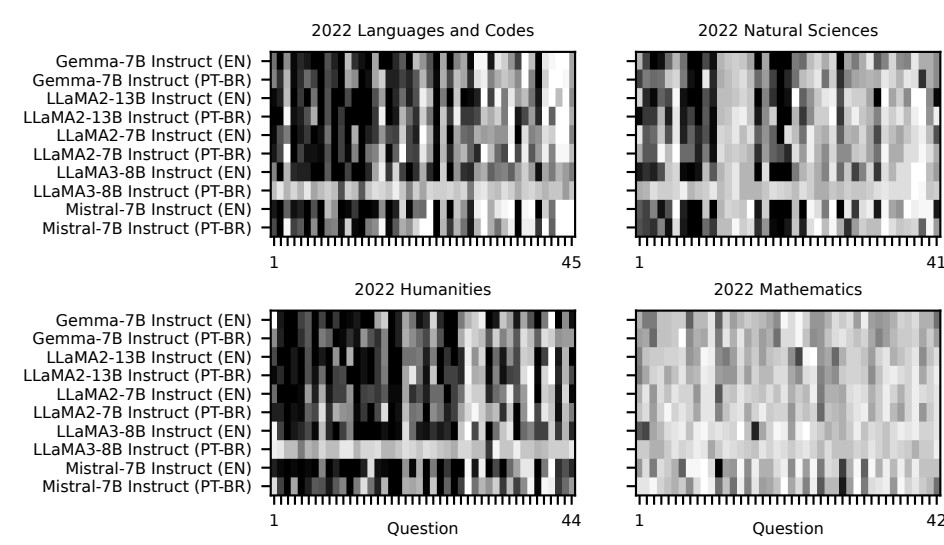

Figure 28: Response patterns for each LLM, where darker indicates more often correct. Questions are sorted by difficulty ($\beta$ value). LLMs are instructed tuned open source models with zero-shot.

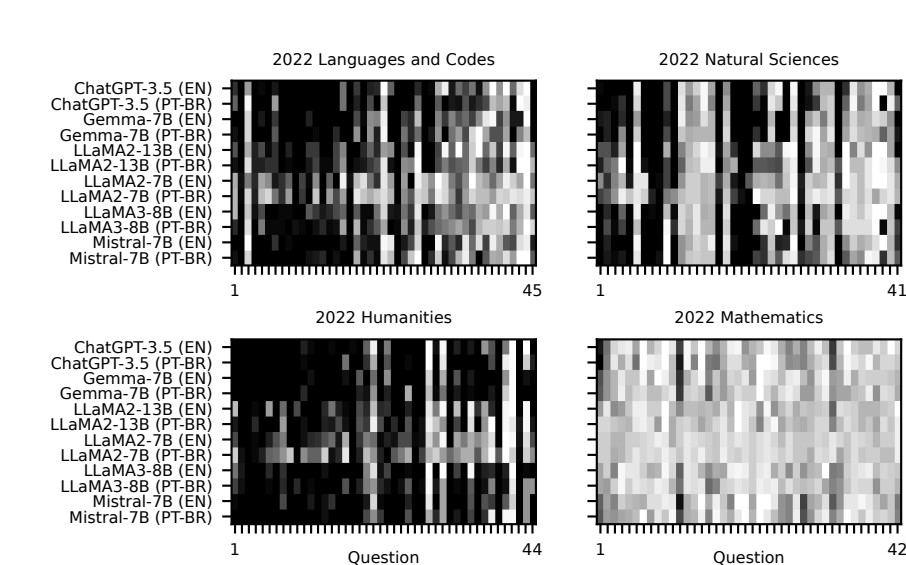

Figure 29: Response patterns for each LLM, where darker indicates more often correct. Questions are sorted by difficulty ($\beta$ value). LLMs are non-instructed tuned open source models and GPT3.5 with one-shot.

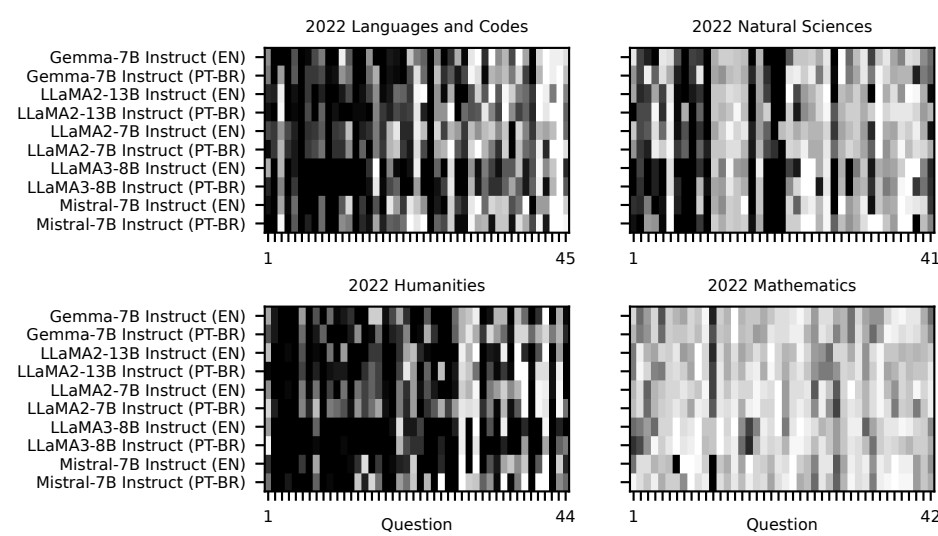

Figure 30: Response patterns for each LLM, where darker indicates more often correct. Questions are sorted by difficulty ($\beta$ value). LLMs are instructed tuned open source models with one-shot.

### A.9  COMPARING IRT $\theta$ AND $l_z$ FOR 2022

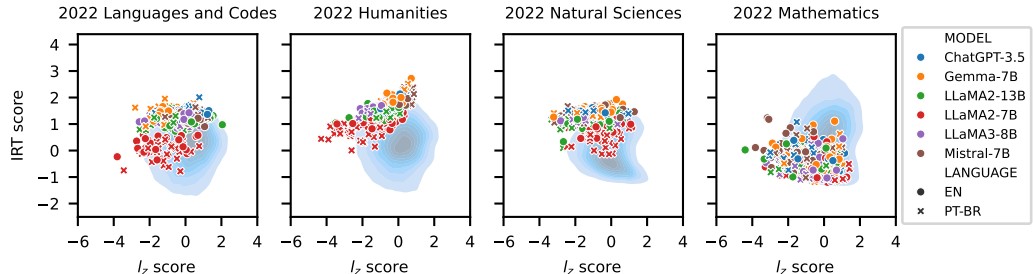

Figure 31: Distribution of $l_z$ and IRT scores for humans and LLMs in the ENEM 2022 exam. LLMs are non-instructed tuned open source models and GPT3.5 with four-shot. LLM datapoints are computed from different shuffles.

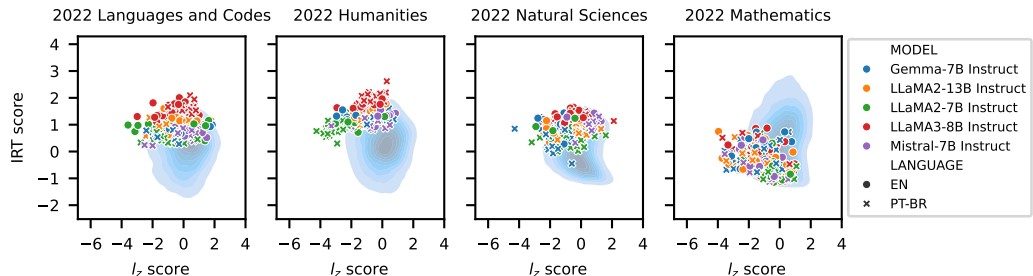

Figure 32: Distribution of $l_z$ and IRT scores for humans and LLMs in the ENEM 2022 exam. LLMs are instructed tuned open source models with four-shot. LLM datapoints are computed from different shuffles.

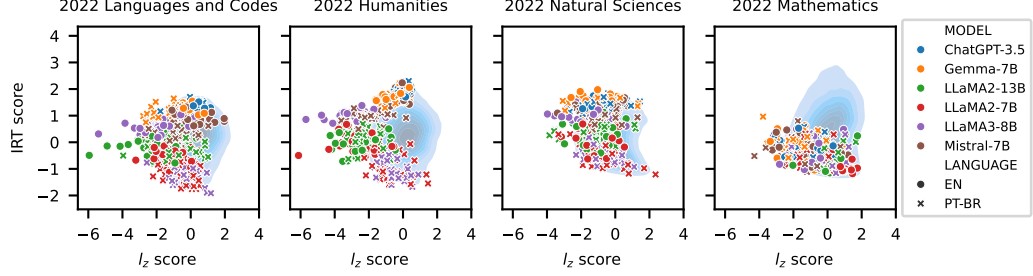

Figure 33: Distribution of $l_z$ and IRT scores for humans and LLMs in the ENEM 2022 exam. LLMs are non-instructed tuned open source models and GPT3.5 with zero-shot. LLM datapoints are computed from different shuffles.

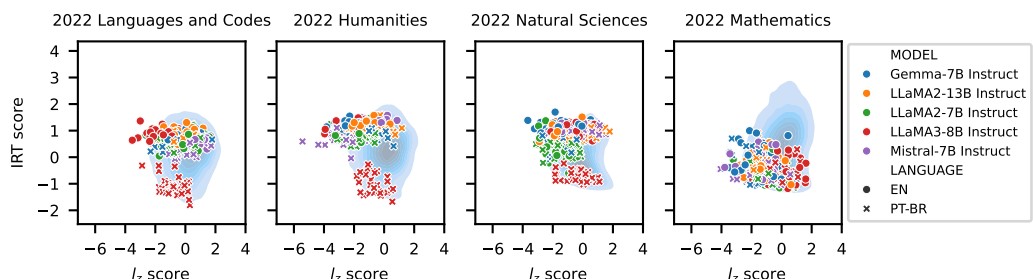

Figure 34: Distribution of $l_z$ and IRT scores for humans and LLMs in the ENEM 2022 exam. LLMs are instructed tuned open source models with zero-shot. LLM datapoints are computed from different shuffles.

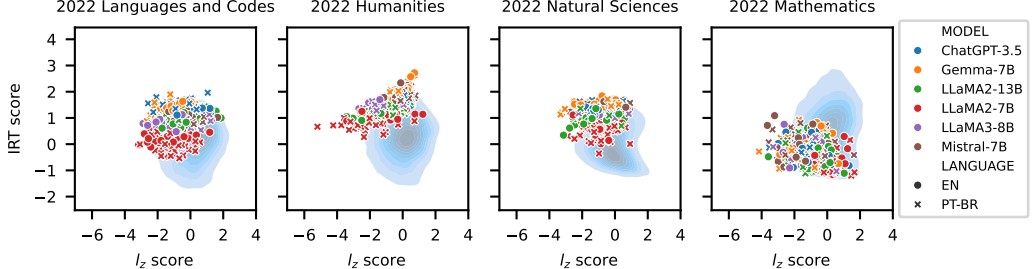

Figure 35: Distribution of $l_z$ and IRT scores for humans and LLMs in the ENEM 2022 exam. LLMs are non-instructed tuned open source models and GPT3.5 with one-shot. LLM datapoints are computed from different shuffles.

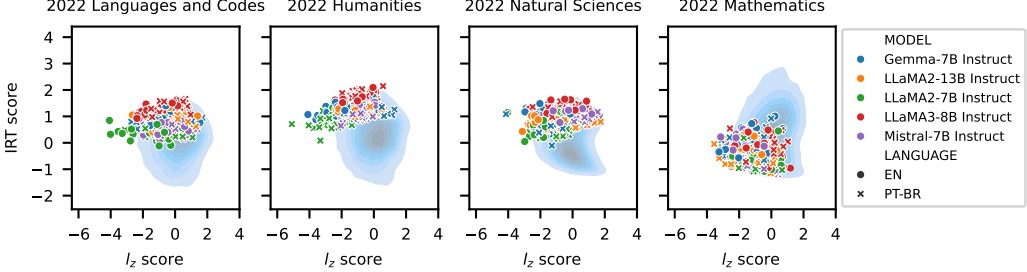

Figure 36: Distribution of $l_z$ and IRT scores for humans and LLMs in the ENEM 2022 exam. LLMs are instructed tuned open source models with one-shot. LLM datapoints are computed from different shuffles.

### A.10 EXAMPLES OF NON-DISCRIMINATING AND HIGHLY DISCRIMINATING ITEMS FOR THE 2023 NATURAL SCIENCES EXAM.

#### A.10.1 POORLY DISCRIMINATIVE QUESTIONS

QUESTION 107 (DISCRIMINATION INDEX -0.013)

Municipalities are responsible for managing their urban waste (garbage) cleaning and collection according to the Federal Constitution. However, there are reports that part of this waste winds up incinerated, releasing toxic substances into the environment and causing explosions-related accidents when incinerating aerosol bottles (e.g., deodorants, insecticides, and repellents). The high temperature causes all the contents inside these bottles to vaporize, increasing the internal pressure until it explodes.

Suppose there is a metal aerosol bottle with a capacity of 100 milliliters containing 0.1 mol of gaseous products at a temperature of 650 degrees Celsius at the moment of explosion.

Consider: $R = \frac{0.082 \times \text{liter} \times \text{atmosphere}}{\text{mol} \times \text{Kelvin}}$

The pressure, in atmospheres, inside the flask at the moment of the explosion is closest to

    A. 756

    B. 533

    C. 76

    D. 53

    E. 13

QUESTION 108 (DISCRIMINATION INDEX -0.076)

The circuit with three identical incandescent light bulbs, shown in the figure, consists of a mixed association of resistors. Each bulb (L1, L2, and L3) is associated in parallel with a resistor of resistance R, forming a set. These sets are connected in series, with all the bulbs having the same brightness when connected to the power supply. After several days in use, only lamp L2 burns out, while the others remain lit.

Figure description: a power supply connected to three sets, arranged in series clockwise, in the following sequence: the parallel set of L1 and R, the parallel set of L2 and R, and the parallel set of L3 and R.

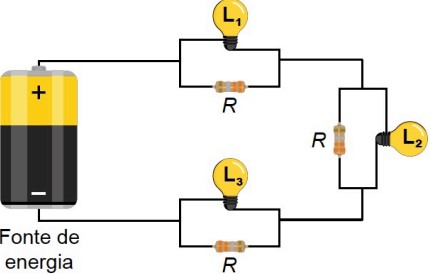

Figure 37: Question 108 Natural Sciences

In the case where all the bulbs work, after L2 burns out, the brightness of the bulbs will be

    A. the same.

    B. more intense.

    C. less intense.

    D. less intense for L1 and the same for L3.

    E. more intense for L1 and less intense for L3.

QUESTION 109 (DISCRIMINATION INDEX 0.013)

A company's transport safety team is evaluating the behavior of the tensions that appear in two horizontal ropes, 1 and 2, used to secure a load of mass M equal to 200 kilograms to the truck, as shown in the illustration. When the truck starts from rest, its acceleration is constant and equal to 3 meters per second squared, while when it arbitrarily brakes, its braking is constant and equal to 5 meters per second squared. In both situations, the load is about to move, and the direction of the truck's movement is shown in the figure. The coefficient of static friction between the box and the bottom surface of the body is 0.2. Consider the acceleration due to gravity to be 10 meters per second squared, the initial tension in the ropes is zero, and the two ropes are ideal.

Figure description: a truck traveling horizontally to the right (represented by the vector V). A box M is resting on the central surface of its body. The box is attached to the rear of the body by horizontal rope 1 and to the front by horizontal rope 2.

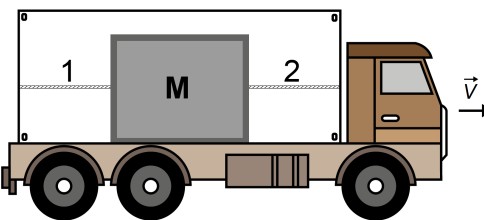

Figure 38: Question 109 Natural Sciences

When the truck is accelerating and braking, the tensions in ropes 1 and 2 in Newton will be

    A. acceleration: T1=0 and T2=200; braking: T1=600 and T2=0.

    B. acceleration: T1=0 and T2=200; braking: T1=1400 and T2=0.

    C. acceleration: T1=0 and T2=600; braking: T1=600 and T2=0.

    D. acceleration: T1=560 and T2=0; braking: T1=0 and T2=960.

    E. acceleration: T1=640 and T2=0; braking: T1=0 and T2=1040.

### A.10.2 HIGHLY DISCRIMINATIVE QUESTIONS

QUESTION 124 (DISCRIMINATION INDEX 0.650)

Update of the Portuguese Society of Neonatology's recommendation

Glass containing aluminum is an excellent material for packaging medicines and supplements because heating can sterilize it. However, when the drug or supplement contains substances that bind strongly to this metal's ion, the aluminum's dissolution is promoted by the displacement of the chemical equilibrium established between the species immobilized in the glass and the species in solution. For this reason, it is recommended that newborn nutrition supplements containing calcium gluconate be packaged in plastic containers rather than in this type of glass.

If this supplement is packaged in this type of glass, the risk of contamination by aluminum will be greater if the

    A. glass of the bottle is translucent.

    B. concentration of calcium gluconate is high.

    C. glass bottle is thicker.

    D. glass is previously sterilized at high temperatures.

    E. reaction of aluminum with calcium gluconate is endothermic.

QUESTION 91 (DISCRIMINATION INDEX 0.624)

It is a common requirement to turn off devices, such as cell phones, whose operation involves emitting or receiving electromagnetic waves when traveling by plane. The justification for this procedure is, among other things, the need to eliminate sources of electromagnetic signals that could interfere with the pilots' radio communications with the control tower.

This interference can only occur if the waves emitted by the cell phone and those received by the plane's radio

    A. are both audible.

    B. have the same power.

    C. have the same frequency.

    D. have the same intensity.

    E. propagate at different speeds.

QUESTION 130 (DISCRIMINATION INDEX 0.621)

The number of bees is in decline in various regions of the world, including Brazil, and multiple factors are contributing to the collapse of their hives. In the United States, seed bombs of native plant species have been used to combat the disappearance of these insects. They are small balls filled with seeds, compost, and clay. When they are thrown and exposed to sun and rain, they germinate even in poorly fertile soil.

This method contributes to the preservation of bees because

    A. it reduces predation.

    B. it reduces the use of pesticides.

    C. it reduces competition for shelter.

    D. it increases the food supply.

    E. it increases breeding sites.

## A.11 DESCRIPTION OF EXAMS

The **Humanities** exam assesses understanding of geographical, cultural, and socioeconomic transformations, as well as comprehension of social and political institutions, technological changes, and the use of historical knowledge to promote conscious engagement in society. It requires recognizing the interactions between society and nature in various historical and geographical contexts.

The **Languages and Codes** exam assesses the use of communication in various contexts. This includes some knowledge and use of foreign languages, understanding of body language, analysis and interpretation of expressive resources in different languages, comprehension of opinions in specific languages, and understanding the impact of communication on personal and social life.

The **Natural Sciences** exam assesses understanding of natural sciences and recognizing their roles in production, economic and social development. It involves associating environmental degradation or conservation with productive and social processes, understanding the interactions between organisms and the environment, and applying specific knowledge of physics, chemistry, and biology.

The **Math** exam assesses the usage of geometric knowledge to represent reality, understanding notions of magnitudes, measurements, and their variations for solving everyday problems, interpreting information of scientific and social nature obtained from reading graphs and tables, and making trend predictions, extrapolations, interpolations, and interpretations.

