# OpenReview forum: "Beyond accuracy: understanding the performance of LLMs on exams designed for humans"
_ICLR.cc/2025/Conference — Submitted to ICLR 2025_

### Official Review · Reviewer_apE4 · 2024-10-31

**Soundness:** 2
**Presentation:** 2
**Contribution:** 2
**Rating:** 3
**Confidence:** 4

**Summary:**

The paper investigates the performance of large language models (LLMs) on human-designed exams, emphasizing the need for deeper analysis beyond standard accuracy metrics. Utilizing a dataset of over 5 million Brazilian students across eight college entrance exams, the authors employ Item Response Theory (IRT) to assess LLM abilities more comprehensively. The study demonstrates that IRT can differentiate between human-like and non-human-like answering patterns and identify specific questions that vary in difficulty for LLMs compared to humans. Ultimately, it argues for the integration of psychometric modeling to better understand LLM capabilities and improve evaluations in academic contexts.

**Strengths:**

1. The study emphasizes the importance of construct validity, highlighting potential limitations of existing exams in measuring LLM abilities, thereby promoting a deeper understanding of LLM performance.
2. By employing IRT, the research provides a more nuanced analysis of LLM performance, distinguishing between human-like and non-human-like response patterns, which leads to more reliable ability assessments.
3. The study leverages a dataset of over 5 million student performances, providing a strong empirical foundation for analyzing LLM behavior, which enhances the credibility of the findings.

**Weaknesses:**

1. The paper relies on a single dataset for evaluating LLMs, which introduces bias. Given the complexity of large models, a more comprehensive evaluation is necessary, making it essential to use a variety of datasets for assessment.
2. The paper exclusively employs the IRT model for assessment. However, there are many other cognitive diagnostic models available that can evaluate learners' abilities, such as MF[1], MIRT[2], and NCD[3]. The authors should explore these alternative models in greater depth to provide a more robust evaluation framework.
3. The paper's technical innovation appears to be limited, primarily focusing on using IRT to evaluate LLMs. The methods employed mainly rely on prompting techniques, which do not demonstrate significant advancements in the evaluation approach.

[1] Andreas Toscher and Michael Jahrer. Collaborative fltering applied to educational data mining. KDD cup, 2010.

[2] Mark D Reckase. Multidimensional item response theory models. In Multidimensional item response theory, pages 79–112. Springer, 2009.

[3] Fei Wang, Qi Liu, Enhong Chen, Zhenya Huang, Yuying Chen, Yu Yin, Zai Huang, and Shijin Wang. Neural cognitive diagnosis for intelligent education systems. In Proceedings of the AAAI Conference on Artifcial Intelligence, pages 6153–6161, 2020.

**Questions:**

1. The study relies solely on Brazil's university entrance exams. Is there a risk of cultural or educational system biases? Can the findings be generalized to LLM evaluations in other countries or educational contexts?
2. There are several models, such as MF and NCD, that can assess students' abilities more effectively than the IRT model. Why did the authors choose to use IRT to evaluate LLMs?

---

### Official Review · Reviewer_NU5S · 2024-11-03

**Soundness:** 2
**Presentation:** 3
**Contribution:** 1
**Rating:** 3
**Confidence:** 4

**Summary:**

The authors suggest comparing IRT parameters of LLMs vs. humans instead of just accuracy. They present results on a large dataset.

**Strengths:**

The paper is well written.

**Weaknesses:**

To me, Figure 1 suggests that accuracy and ability estimates and highly correlated.

This approach seems to have been already studied by: Liu, Yunting, Shreya Bhandari, and Zachary A. Pardos. "Leveraging LLM-Respondents for Item Evaluation: a Psychometric Analysis." arXiv preprint arXiv:2407.10899 (2024). I agree that this paper is recent, though.

The approach of using IRT for making more efficient benchmarks seems to be taken by Polo et al. (2024) and Zhuang et al. (2023), papers cited by the authors. However I do not feel that considering a IRT model trained on human responses, as stated by the authors, can be considered enough novel. Plus, the way the estimation of IRT parameters (LLM ability estimates) is done (if it depends on a prior, then it is biased) can hinder the reproducibility of results and the validity of findings.

**Questions:**

Shouldn't we estimate multidimensional IRT parameters of models vs. humans instead of just 2PL-IRT?

---

### Official Review · Reviewer_vQrG · 2024-11-04

**Soundness:** 1
**Presentation:** 3
**Contribution:** 1
**Rating:** 3
**Confidence:** 5

**Summary:**

The paper examines whether LLMs demonstrate human-like reasoning on exams designed for humans by using Item Response Theory (IRT). Analyzing a dataset of over 5 million Brazilian students' responses to college entrance exams, the study finds that traditional accuracy metrics inadequately assess LLM capabilities. IRT, by accounting for question difficulty, offers a more nuanced evaluation, distinguishing between human-like and non-human-like response patterns. The results show that while LLMs sometimes mimic human behavior, they often deviate significantly.

**Strengths:**

1. The paper is well-organized and clearly articulates both the limitations of using accuracy as the sole metric and the benefits of IRT for evaluation. Diagrams and data tables effectively support the paper’s arguments, making complex psychometric methods accessible to a broader audience.
2. The paper employs rigorous experimental design and utilizes a comprehensive dataset, enhancing the robustness of its findings. By analyzing various models, including GPT-3.5 and LLaMA variants, the authors demonstrate the generalizability of IRT’s applicability. The study further uses well-defined psychometric to validate its claims, supporting the soundness of the technical approach.
3. This work holds significance as it points out weaknesses in LLM evaluations. By moving beyond accuracy, the paper demonstrates that psychometric techniques can better represent model abilities quantitatively.

**Weaknesses:**

The paper has several notable shortcomings.
1. Firstly, the idea of using psychometrics and IRT to replace traditional metrics like accuracy in AI benchmarking was proposed well before 2021, diminishing the novelty of the approach.
2. The use of IRT to compare the response patterns of LLMs with those of humans has already been widely explored in existing research.
3. The technical methods employed in the paper, such as IRT and Fisher information maximization, are already extensively applied in AI evaluation, further reducing the originality of the study's methodology.

Presentation needs to be polished , and it remains some typos.
Results section, line 271 , “LMM” may be a typo
Methods section, line 237, “run” -> ”ran”

**Questions:**

Please emphasize the innovative aspects and contributions of the paper.

---

### Meta-Review · Area_Chair_3tgn · 2024-12-11

**Metareview:**

This paper is appreciated for providing a new perspective on evaluating LLMs using IRT; however, its approach does not significantly go beyond the scope of existing studies. In addition, concerns have been raised about dataset biases and the lack of consideration for alternative models, which limits the contribution and technical originality.

**Additional Comments On Reviewer Discussion:**

Nothing

---

### Decision · Program_Chairs · 2025-01-22

Reject